# Dakar Niño under global warming investigated by a high-resolution regionally coupled model

Shunya Koseki[1], Rubén Vázquez[2,3], William Cabos[2], Claudia Gutiérrez[2], Dmitry V. Sein[4,5], Marie-Lou Bachèlery[1,6]

[1]Geophysical Institute, University of Bergen / Bjerknes Centre for Climate Research, Bergen, 5007, Norway
[2]Departamento de Física y Matemátics,Universidad de Alcalá, Alcalá de Henares, 28801, Spain
[3]Instituto Universitario de Investigación Marina (INMAR), Universidad de Cádiz, Cádiz, 11510, Spain
[4]Alfred-Wegener Institute for Polar and Marine Research, Bremerhaven, 27570, Germany
[5]Shirshov Institute of Oceanography, Russian Academy of Science, Moscow, 117218, Russia
[6]Climate Simulation and Prediction Division, Euro-Mediterranean Center on Climate Change, Bologna, 40127, Italy

*Correspondence to*: Shunya Koseki (Shunya.Koseki@uib.no)

**Abstract.** In this study, we investigated the interannual variability of sea surface temperature (SST) along the northwest African coast, focusing on the strong Dakar Niño and Niña events, and their potential alterations under the RCP8.5 emission scenario of global warming using a high-resolution regional coupled model. Our model accurately reproduces the SST seasonal
cycle along the northwest African coast and its interannual variability in terms of amplitude, timing, and position of the maximum variability. Comparing the Dakar Niño variability between the 1980-2010 and 2069-2099 periods, we found that its variability intensifies under a warmer climate without changing its location and timing. The intensification is more pronounced during Dakar Niñas (cold SST events) than during Niños (warm SST events). In the future, SST variability is correlated with ocean temperature and vertical motion at deeper layers. The increase of Dakar Niño variability can be explained by the larger
variability in meridional wind stresses, which is likely to be amplified in the future by enhanced land-sea thermal contrast and associated sea-level pressure anomalies elongated from the Iberian-Mediterranean area. A heat budget analysis in the mixed-layer suggests that the surface heat flux and horizontal advection anomalies are comparably important for Dakar Niño/Niña in the present climate. However, the future intensification of the Dakar Niños/Niñas is likely to be driven by the surface heat flux (latent heat flux and shortwave radiation). While horizontal and vertical advection anomalies also contribution to the
intensification, their roles are secondary.

## 1 Introduction

Climatologically, the Senegal-Mauritania Frontal Zone (SMFZ, around 9°N-14°N and 20°W-16°W) is one of the most pronounced oceanic frontal zones generated along the eastern boundary current system (Oettil et al., 2021 and Fig.1b). The cold water of the southward flowing Canary Current and Senegal-Mauritania Upwelling System (Barton et al., 1998;
Perez-Hernandez et al., 2013; Vazquez et al., 2022) meet the relatively warm tropical water, creating a steep sea surface temperature (SST) gradient (Ndoye et al., 2014; Sylla et al., 2019). The northern boundary of the SMFZ is around 19°N where

the Canary Current joints with the north equatorial Current (e.g., Santana-Falcon et al., 2020) around Cape Blanc (e.g., Pastor et al., 2008). The Canary upwelling system is tightly connected with the equatorward alongshore wind associated with the Azores anti-cyclone (e.g., Davis et al., 1997) and highly influenced by the latitudinal migration of the Intertropical Convergence Zone (ITCZ, Sylla et al., 2019). Due to the enriched nutrients from the ocean subsurface, the SMFZ and Canary upwelling region also feature an active marine ecosystem (e.g., Aristegui et al., 2009; Gomez-Letona et al., 2017), playing an important role in local and regional fisheries such as sardinella from the northwestern Africa to the Iberian coasts in the north tropical-to-subtropical Atlantic (e.g., Arrasate-Lopez et al., 2012; Becognee et al., 2006; Ndoye et al., 2014).

Apart from these climatological mean-state features, the SMFZ shows intense interannual variability in SST (shown in Fig.1b) with extreme warm anomalies know as Dakar Niño (Oettli et al., 2016). Dakar Niño is primarily associated with the local wind anomaly and it peaks between March and April and surface heat flux plays a crucial role in its development (Oettli et al., 2016). A similar mode of SST variability is found in the southeastern tropical Atlantic, known as Benguela Niño (Bachelery et al., 2020; Koungue et al., 2021; Koungue et al., 2019; Rouault et al., 2018). There, the interannual variability is driven not only by local wind fluctuations, but is also strongly linked to western equatorial winds that trigger the propagation of equatorial Kelvin waves and coastal trapped waves off the African coast (Bachelery et al., 2020; Koungue et al., 2021; Koungue et al., 2019; Rouault et al., 2018). The inter-annual SST variability in the Dakar system has a major influence on marine ecosystems. For instance, Lopez-Parages et al. (2020) showed that the distribution of round sardinella tends to be modified following the Dakar-Niño-like pattern initialized by El Niño variability in the tropical Pacific.

For sustainable development, including the fisheries sector, the understanding of climate variability under global warming draws increasingly attention not only from the scientific community, but also from societies, stakeholders, and governments. Climate projections from Earth System Model (ESM), such as the Coupled Model Intercomparison Project Phase6 (CMIP6; Eyring et al., 2016), are one of the most common tools to investigate future climate change. These ESMs are state-of-the-art models that have been improved in many aspects for the simulation of the climate system and their use for climate prediction (e.g., Bracegirdle et al., 2020; Priestley et al., 2020; Choudhury et al., 2022). However, model biases in the tropical Atlantic climate are a long-standing issue even in CMIP6 and are very common in most of state-of-the-art ESMs (Richter and Xie, 2008; Cabos et al., 2017; Voldoire et al., 2019; Richter and Tokinaga, 2020). These biases are one of the main sources of uncertainty in climate projections and therefore, there is a necessity to utilize ESMs with fewer systematic baises to assess more plausible climate projections. Partially due to the model errors mentioned above and relatively recent discovery of Dakar Niño (the first paper on this topic is Oettli et al., 2016), there are few studies on how the Dakar Niño variability would evolve under global warming while studies on the equatorial Atlantic variability have been reported recently (Crespo et al., 2022; Yang et al., 2022).

Several methodologies have been proposed in the previous studies to alleviate model errors, including the implementation of better parameterization (e.g., Deppenmeier et al., 2020), heat and/or momentum flux correction/anomaly coupling (e.g., Dippe et al., 2018; Toniazzo and Koseki, 2018; Voldoire et al., 2019), and interactive model ensembles (e.g., Shen et al., 2016; Counillon et al., 2023; Schevenhoven et al., 2023). Apart from these methodologies, resolution refinement

is also beneficial to improve the model performance in the tropical Atlantic (e.g., De La Vara et al., 2020). However, Sylla et al. (2022) by assessing the archives of High Resolution Model Intercomparison Project (HighResMIP, Haarsma et al., 2016), stressed the limited benefits of model refinement to improve the Canary Current upwelling system. On the other hand, Vazquez et al. (2022) suggest that a high-resolution (mesoscale eddy-permitting scale) regional coupled model is capable of accurately
representing the Canary Current upwelling systems and surface wind field.

This study, therefore, aims to unveil how the Dakar Niño variability might change in the future climate using the reliable high-resolution regional coupled model used in Vazquez et al. (2022). This paper is structured as follows: Section 2 gives details on the regional coupled model, the experimental setup, and the reanalysis data. We will present the results of model simulations with brief evaluation comparing with reanalysis data in Section 3. In Section 4, we offer discussions on the
processes that can change Dakar Niño employing a heat budget analysis following Oettli et al. (2016). The details of heat flux budget is given in Section 4.2. Finally we will summarize the study findings in Section 4.3.

## 2 High-resolution regional coupled model

The regionally-coupled model ROM (e.g., Sein et al., 2015; Sein et al., 2020) configurations used in this study are the same as in Vazquez et al. (2022). It consists of a regional atmospheric component, namely limited-area Regional Model
(REMO; e.g., Jacob, 2001) and global oceanic component, which is the Max-Planck Institute Ocean Model (MPIOM, e.g., Marsland et al., 2003; Jungclaus et al., 2013). REMO has 25km horizontal resolution with 27 hybrid vertical levels. MPIOM adapts an orthogonal curvilinear horizontal grid system with shifted poles allowing to refine the focused region while a global domain can be maintained (for more details, see Sein et al., 2015). In our setting, MPIOM has 5 to 10km of horizontal resolution around the Iberian Peninsula and Cape Ghir at 31°N and 10°W upscaling gradually toward 100km in the Southern Ocean. The
ROM's configuration domain utilized in this study is given in Fig.1. Air-sea coupling between REMO and MPIOM is active within the yellow rectangular shown in Fig.1. Outside of the active regional coupling, the MPIOM is forced by prescribed atmospheric forcing, while REMO is laterally forced by the same prescribed atmospheric forcing.

In this study, ROM is integrated from 1950 to 2099 under both historical conditions and the Representative Concentration Pathway 8.5 (RCP8.5) forcing where the anthropogenic emission of greenhouse gases increases until the end of
the century. The global atmospheric forcing is derived from the low-resolution Max Planck Institute ESM (MPI-ESM-LR, Block and Mauritsen, 2013; Giorgetta et al., 2013). At detailed evaluation of the ROM configurations for 1950-2005 historical period, using observational products and forced by ERA-Interim (Dee et al., 2011) is extensively demonstrated by Cabos et al. (2020), Cabos et al. (2017), and Vazquez et al. (2022). Here, we analyze the data from 1980 to 2010 as historical climate conditions and from 2069 to 2099 as future climate change referring to them as $ROM_P$ and $ROM_F$, hereafter. For a brief
evaluation of the ROM simulation, atmosphere and ocean reanalysis data provided by the European Centre for Medium Range Weather Forecast ERA5 (Hersbach et al., 2020) and ORAS5 (Zuo et al., 2019), during 1980-2010, along with the satellite data of European Space Agency (ESA) SST Climate Change Initiative (CCI) product (Good, 2019) for 1981-2010 is used.

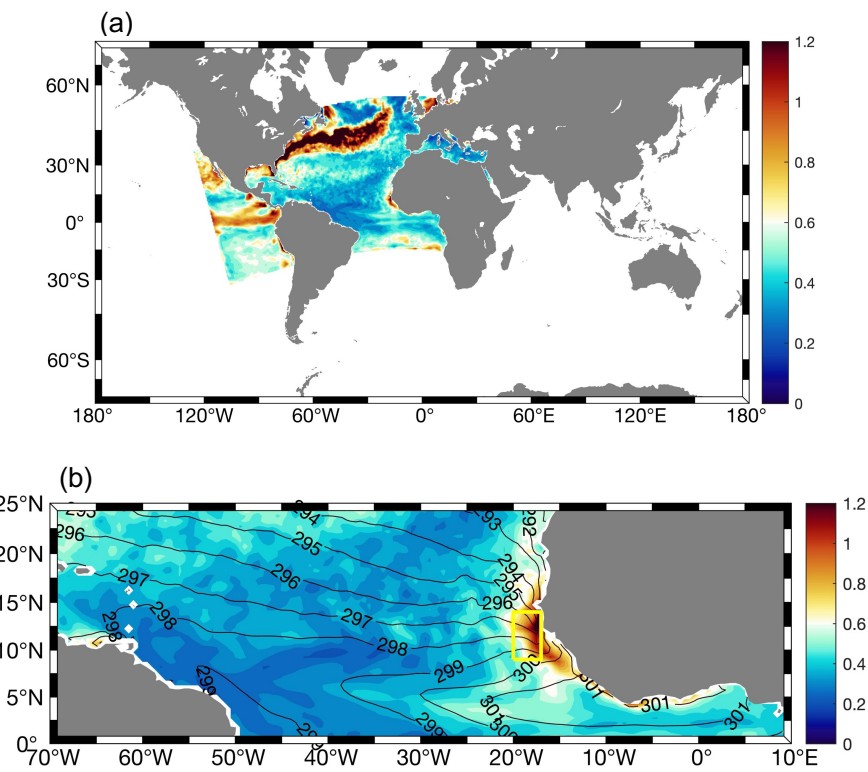

**Figure 1: (a)** Schematic of the ROM's domain used in this study. The area where the atmosphere and ocean are coupled is highlighted in color, which represents the SST standard deviation (K) of $ROM_P$ simulation in March. (b) a zoomed-in of panal (a) with SST climatology (K) of $ROM_P$ in March shown with contours. The yellow box indicates the Dakar Index region.

## 3 Results

### 3.1 Climatology and interannual variability

First, we assess the SMFZ seasonal cycle and its interannual variability as shown in Fig.2. The results show a clear seasonal cycle displacement of the SMFZ with the cold water penetrating further southward from February to April and being pushed further northward from August to October (Fig. 2a-d). This seasonal meridional migration of the SST front is linked to the seasonal cycle of the Canary upwelling system (e.g., Cropper et al., 2014; Pardo et al., 2011; Sylla et al., 2019) and associated with the northward migration of the ITCZ in the summer months, which displaces the surface water massed

meridionally. Additionally, during winter to early spring, the Mauritania Current flows southward to around 14°N. Inversely, associated with the relaxation of the trade winds (e.g., Lazaro et al., 2005), the Mauritania Current shifts northward to reach the Cape Blanc (around 20°N), which is associated with the cessation of upwelling south of this latitude (Mittelstaedt, 1991). This results in a 2.5-fold increase in northward flow during summer compared to the upwelling season, transporting waters of mainly South Atlantic origin into the SMFZ (Klenz et al., 2018). The steep SST gradient is consistent with the SST seasonal

cycle and locates at 10°N-12°N in February to April and 20°N-22°N in August to October. Coinciding with the position of the front, enhanced interannual variability appears in November and persists till May with a maximum peak of 1.2 K at 10°N-12°N between February and April (Fig. 2e). This period coincides with the preferred season of Dakar Niño/Niña (Oettli et al., 2016). Another moderate peak of variability is found from August to October at 20°N-22°N when the SST gradient reaches its second maximum. Similar to these patterns of the SMFZ and Dakar Niño, the Angola-Benguela Frontal Zone (ABFZ; e.g, Colberg and Reason, 2006; Koseki et al., 2019) and Benguela Niño variability also peaks between February to April (e.g., Aristegui et al., 2009; Rouault et al., 2018; Koungue et al., 2019; Bachelery et al., 2020; Koseki and Koungue, 2021; Koungue et al., 2021;). However, there are dissimilarities between the two coastal interannual modes: the seasonal displacement of the SMFZ is significantly wider than the ABFZ whose position is almost seasonally fixed (e.g., Koseki et al., 2019). The ESA SST shows a similar pattern of seasonality of SST, variability and SST gradient (Figs. 2b and f). Compared to the ERA5, the ESA SST is cooler in all months. This discrepantly could be due to relatively-poor representation of coastal upwelling in the ERA5 (which is coarser than ESA) and the fact that ERA5 has a warm bias (Vazquez et al., 2022). Conversely, the SST meridional gradient is much steeper in the ESA than the ERA5, likely because the ESA has a finer resolution (0.05 degree) than the ERA5 (0.25 degree).

The ROM$_P$ simulation accurately reproduces the SMFZ well as shown in Figs. 2c compared to Fig 2a. At the lower latitudes (EQ to 12°N), ROM$_P$ has cold SST biases during the whole year with respect to the ERA5 and ESA. Such cold SST bias can be also seen at the higher latitudes (18°N to 30°N), but they are more moderate (Fig. 2b). According to Vázquez et al. (2023), coupling and higher-resolution SST enhance the representation of the North African Coastal Low Level Jet (Soares et al., 2019), which is a key feature of the surface wind field along the North African coast. This accurate representation of the SMFZ is due to the finer resolution, which permits meso-scale eddy and filaments in our focus region in contrast to the common coupled climate models like CMIP5 (Vázquez et al., 2022). SST variability is also realistically represented in ROM$_P$ (Fig. 2g). The variability is maximized during March to April, which is slightly delayed from the observation. However, its amplitude is as strong as the ERA5 but weaker than ESA (Figs. 2e, f and g). The secondary peak during August to October is also well-captured.

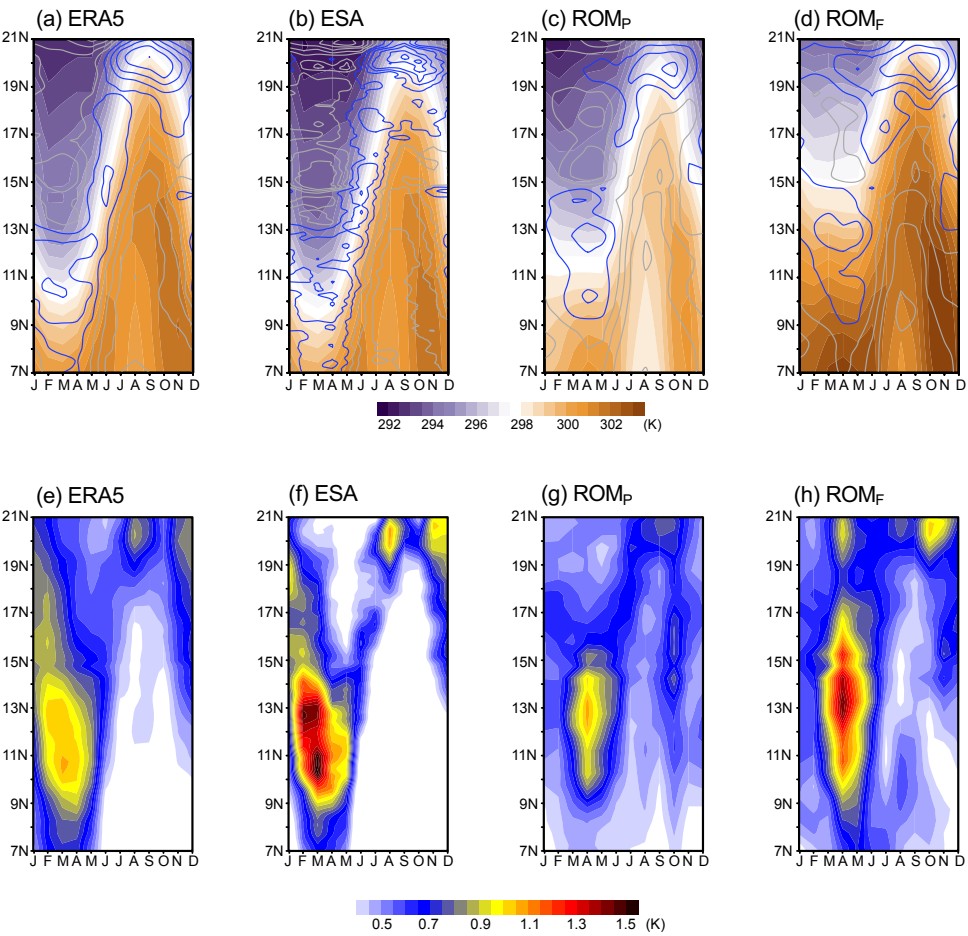

**Figure 2:** Hovmöller plot of (top) sea surface temperature (SST, color) and absolute value of merdional SST gradient (K/100km, contour, interval is 0.2K/100km). The meridional SST gradient greater than 0.5K/100km is shown by blue. Data are averaged between 21°W and 17°W for ERA5, ESA, ROM$_P$ and ROM$_F$, respectively. (bottom) Same as top panels, but for the standard deviation of detrended SST. Unit is in Kelvin.

Under the highest emission scenario, this region experiences significant warming: 3°C in the SMFZ and 1°C at higher latitudes (Fig. 2d). However, the SMFZ location is almost identical between ROM$_P$ and ROM$_F$ (not shown). Interestingly, the Dakar Niño variability is strengthened in both peaks in ROM$_F$ while its timing does not change (Figs. 2h and S1). This response contrasts with the recent studies on the equatorial Atlantic variability (Crespo et al., 2022; Yang et al., 2022). The possible mechanism for this enhancement will be discussed in the next subsection. This study will focus on the month of March as it has been determined to be the peak month of the event. Note that despite the slight differences in timing of the SST variability

peak compared to observations the simulated March variability is comparably intense (Fig. 2). Therefore, we will focus on March throughout the rest of the paper (Fig. S1 gives a time series of SST standard deviation averaged over 9°N-14°N).

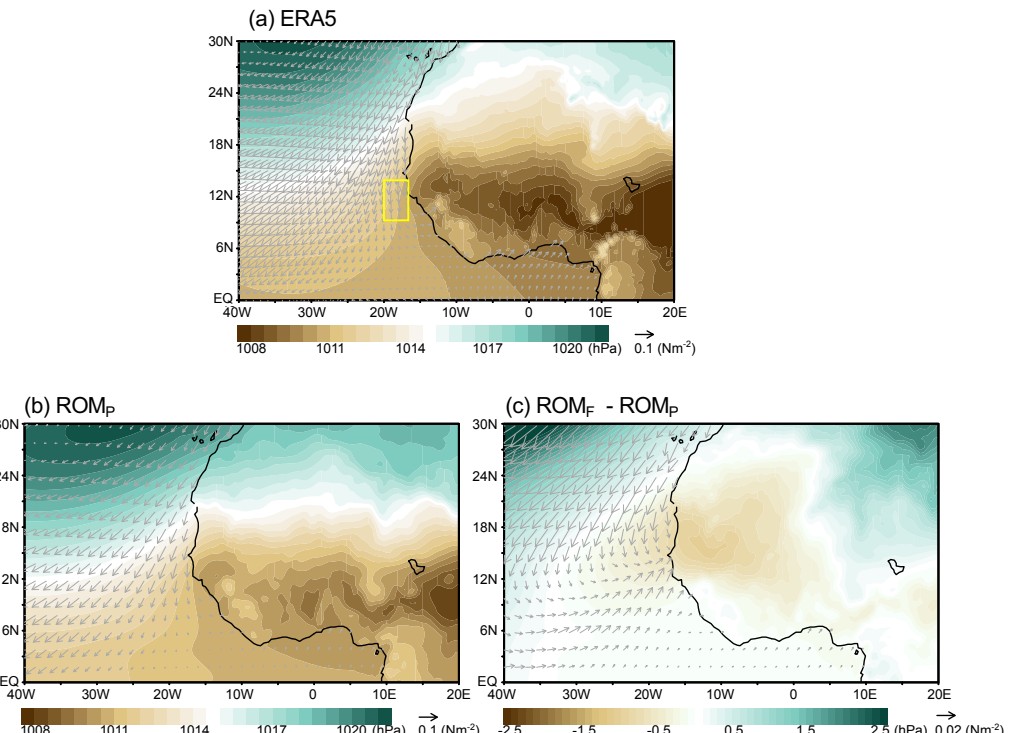

**Figure 3:** March-climatological sea level pressure (SLP in hPa, color) and wind stress (arrows) in (a) ERA5 and (b) ROM$_P$. (c) The difference in SLP and wind stress climatology in March between ROM$_F$ and ROM$_P$. The yellow box indicates the Dakar Index region.

As shown in Fig. S2, associated with the intense upwelling, the thermocline (20°C isotherm) tilts zonally (shallower in the east) in the reanalysis (Fig. S2a). ROM$_P$ can represent this zonal tilting of thermocline well with a steep vertical gradient found around 40-60m depth along the coast (Fig. S2b). Under global warming, the thermocline tends to be deeper while the coastal vertical gradient seems stronger than that in ROM$_P$ between 40 and 60m depth (Fig.S2b and c).

In March, a low pressure system dominates over western Africa between 6°N and 15°N, while the Azores high pressure system sits over the North Atlantic. Due to this contrast of surface pressure, strong southerly winds blow along the western African coast (Fig. 3a). ROM$_P$ simulates this atmospheric circulation realistically although the low pressure over the Sahel is slightly underestimated (Fig. 3b). In the future, the continental low pressure is partially deepened, especially near the coastal area (10°N-24°N and 15°E to 0°E as shown in Fig. 3c) where the surface temperature at 2m is intensively warmed by 5 degrees in ROM$_F$ (not shown). This strong terrestrial warming can be explained by the desert amplification (Cook and Vizy, 2015; Zhou, 2016). Corresponding to this deepened low pressure, a cyclonic circulation anomaly is detected around 15°N and 15°W in Fig. 3c. This anomaly pattern is similar to climate projections by CMIP5 (Sylla et al., 2019). While upwelling-favourable wind intensifies at higher latitudes (18°N-30°N), onshore wind anomalies form at lower latitude (12°N-15°N).

### 3.2 Dakar Niño

In this subsection, we investigate the modification of Dakar Niños/Niñas under the highest emission scenario employing lag-correlation and composite analyses. These analyses are based on the March Dakar Index, defined as detrended interannual SST anomalies averaged over the 21°W-17°W, 9°N-14°N box (Oettli et al., 2016). Dakar Niño and Niña events
are identified by March Dakar Index anomalies that exceed or fall below ±1 standard deviation of the mean Dakar Index. As shown in Fig. 4a, in ERA5 there are 7 Dakar Niño and 6 Dakar Niña events over 31 years. This result of event detection is consistent with that of Oettli et al. (2016) while our study utilizes ERA5 and they employed HadISST.

In comparison, our ROM simulation, $ROM_P$ and $ROM_F$, have 7/9 Dakar Niño and 8/6 Dakar Niña events for the present and future climate, respectively. Note that there is no possible consistency in the timing of Dakar Niño and Niña events
between ERA5 and $ROM_P$ as the $ROM_P$ simulation is a historical run for 31 years. However, the frequency of the events is similar. Under global warming, the frequency of the events seems not strongly influenced (Fig. 4c) although negative events are stronger than in the present climate (Fig. 4b and c). As shown in Fig. S1, the standard deviation of SST in $ROM_F$ intensifies from March to May.

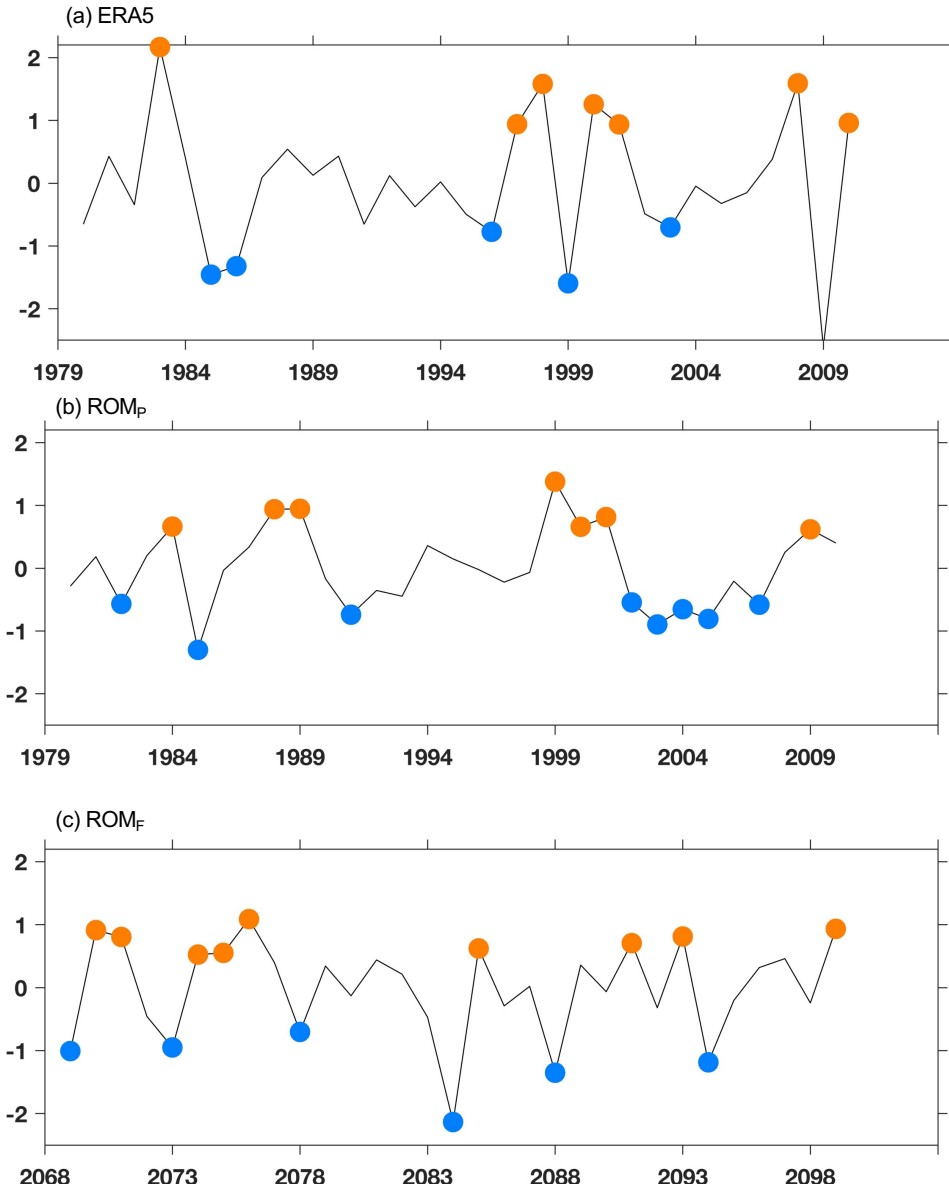

**Figure 4**: Time series of Dakar Index (detrended SST averaged 9°N-14°N and 20°W-17°W) for (a) ERA5, (b) $ROM_P$ and (c) $ROM_F$. Orange and blue dots indicate Dakar Niño and Niña events, respectively.

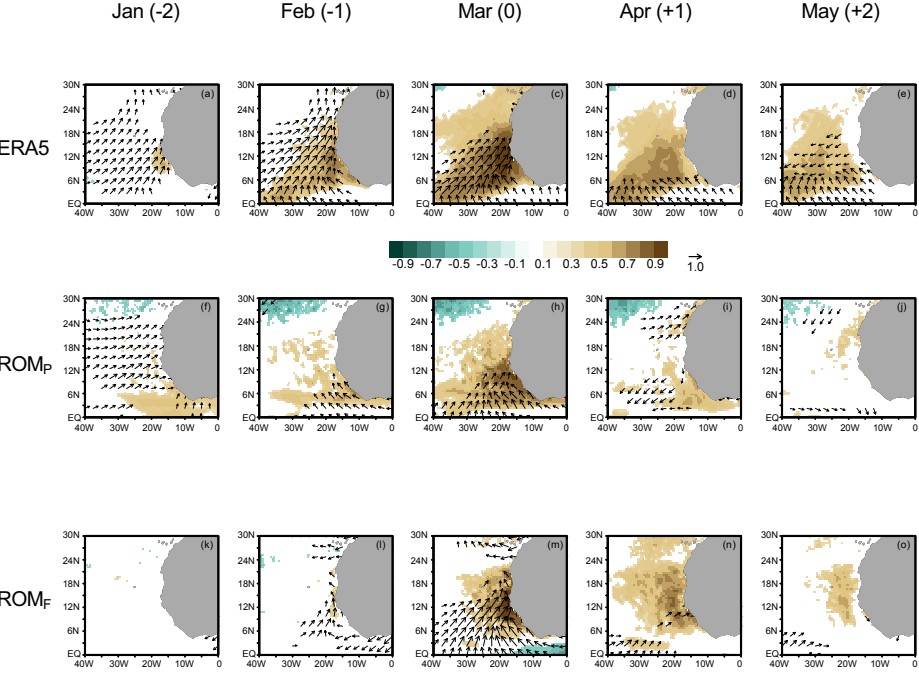

**Figure 5:** Lag-correlation plots between March Dakar Index (SST over 21°W-17°W, 9°N-14°N) and wind stress (vector) and SST (color). Only the correlation is shown satisfying $p < 0.05$ for (top) ERA5, (middle) $ROM_P$, and (bottom) $ROM_F$, respectively. The vector with significant correlation of zonal or meridional wind component is shown. From the left to the right, the panels show the lag-correlation from January (-2) to May (+2).

As shown by Oettli et al. (2016) using a reanalysis data of GODAS, Dakar Niños are strongly correlated with regional coastal winds variability. We also investigate this relationship using surface wind stress and SST from ERA5 and ROM simulations. In ERA5, positive surface wind correlation (southwesterly) and SST positive correlation are found along the west African coast. In January, SST anomalies from ERA5 averaged over the Dakar Niño box are positively correlated with SSTs along the west African coast as well as southwesterly surface wind anomalies (Fig.5a). The correlation strengthens in March which is the peak of the event. This patter indicates that northeasterly winds are reduced in Dakar Niños (Fig.5c). In March, the significant correlation of surface wind is localized south of 15°N. After the peak of Dakar Niños, the surface wind correlation is pronounced only around the equator and offshore in April to May (Figs. 5d and e). The area positively correlated with SST shifts westward, in particular around 6°N to 10°N, in April to May (Figs. 5d and e) and this might be related to Rossby wave propagation, which can influence the equatorial Atlantic Zonal mode in summer (e.g., (Martin-Rey and Lazar, 2019). In $ROM_P$, the life-cycle of Dakar Niño variability is to some extent simulated realistically and the surface wind is significantly correlated in January (Fig. 5f). Positive SST anomalies develop from January to March while the connection between the Dakar Niño index and the wind field in February is not well simulated (Figs. 5f-h). The surface wind in March is more locally correlated compared to ERA5 (Figs. 5c and h). After the peak, the positive SST correlation decays, but seems not

to propagate westward clearly. However, a signal of westward propagation can be detected at 41m depth (around 6°N) in ROM$_P$ (Fig. S3). Although the evolution of the ROM$_F$-simulated Dakar Niño and correlated surface wind in January and February are not as clear as ROM$_P$ (Figs. 5k and l), the surface wind is correlated more broadly along the coast during March up to 18°N (Fig. 5m) while it is limited to 12°N in ROM$_P$ (Fig. 5h). After the peak, the positive SST correlation moves westward more clearly like ERA5 even though its phase speed is slower than the ERA5 (Figs. 5n and o).

As illustrated in Fig.5, ROM simulations show a tight connection between SST and surface winds interannual variability. Along the western African coastal region, the thermocline is shallow due to the wind-driven coastal Ekman divergence (Figs. S2 and S4). Consequently, differences in vertical structure of temperature and vertical motion variability between the present and future climate are also expected. Figure 6 shows the correlation between the Dakar Index and the interannual temperature and vertical velocity anomalies off the coast of SMFZ in ROM simulation during March. In ROM$_P$, the significant positive correlation (0.8 to 0.9) concentrates between the surface and 40m depth and decreases to 0.4 100m depth, which decreases to 0.4 at 100m (Fig. 6a). In contrast, the ocean temperature in ROM$_F$ shows a significant correlation (0.5) with the Dakar Index deeper in the water column down to 160m depth (Fig. 6c). Similar results are also obtained in the vertical motion anomalies (Fig. 6b and d). Negative correlations of vertical motion remains stronger and deeper in ROM$_F$ (significant till 80m) than ROM$_P$ (significant till 40m). This indicate that Dakar Niño and Niña will have a deeper signature in the future.

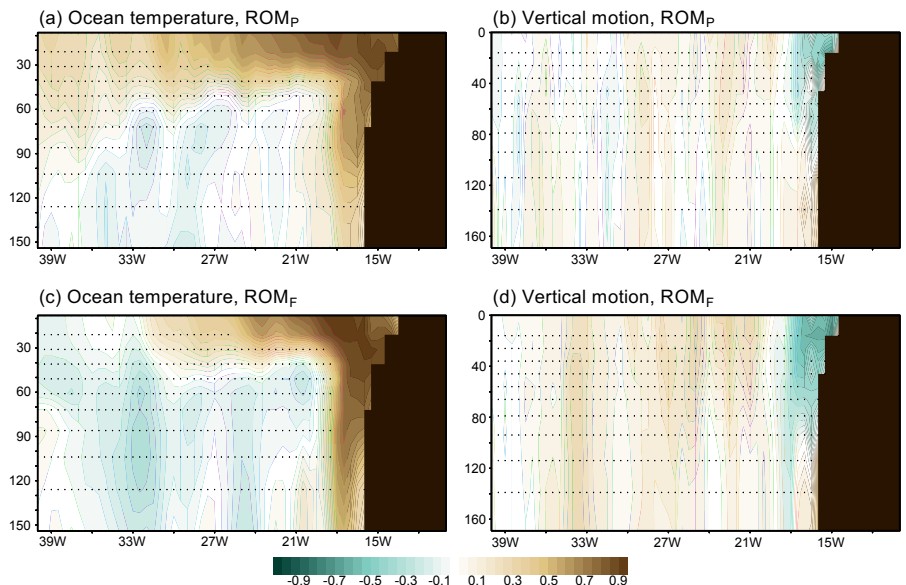

**Figure 6:** Vertical-longitudinal section of the correlation between the Dakar Index and (left) ocean temperature and (right) vertical motion averaged between 9°N and 14°N for (top) ROM$_P$ and (bottom) ROM$_F$. The dots denote no significance of correlation.

Correlation analysis just provides coherency between targeted variables with regardless of their signs. Since Oettli et al. (2016) showed some symmetric features (e.g., magnitude of warm and cold events), we now compare the vertical ocean structure during Dakar Niño and Niña events in the present and future climate. Figures 7a-d show the composite of ocean temperature anomalies during Dakar Niño and Niña in ROM$_P$ and ROM$_F$, respectively. Similar to the correlation plot (Fig. 6), the temperature anomalies in ROM$_P$ are large around 40m depth in both Niño and Niña and their magnitudes are almost identical (±1.8K, Figs. 7a and b). Interestingly, the temperature anomalies in ROM$_F$ around 40 m depth are more pronounced during the Dakar Niñas than Niños (Figs. 7c and d). In addition, the temperature anomaly associated with the Dakar Niñas is detected more deeply in ROM$_F$ than in ROM$_P$ (Figs. 7b and d). That is, the amplification of the variability under global warming is mainly induced by the Dakar Niñas in our simulation. While the ocean experiences overall warming from the surface to subsurface due to climate change, this warming is not uniform (Fig.7e). The ocean surface warms more efficiently than the sub-surface. The difference in warming is particularly large around 40m and upper levels where the temperature anomalies due to Dakar Niño/Niña variability and its change is the most intense (Fig. 7a-d). In addition to the vertical motion change, this strengthened stratification at 40m depth could be a factor in the strengthened Dakar Niño/Niña variability.

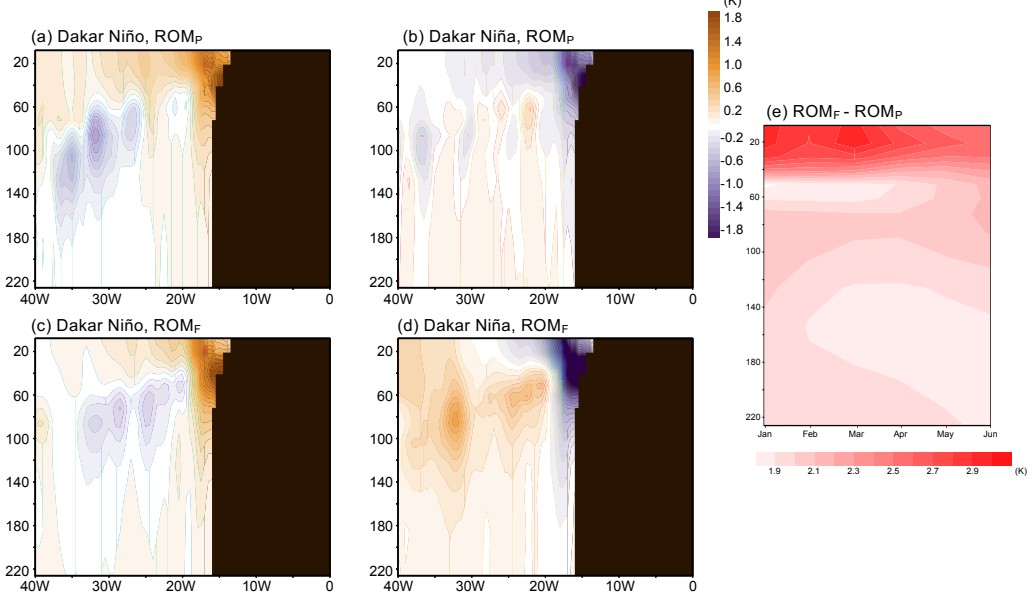

**Figure 7:** (a)-(d) Composite vertical-longitudal section of the temperature anomalies (K) for Dakar Niño/Niña in ROM$_P$ and ROM$_F$ in March. (e) Vertical-temporal section of the monthly climatological ocean temperature difference between ROM$_F$ and ROM$_P$ averaged over 9°N-14°N and 20°W-16°W.

## 4.  Discussion and Summary

### 4.1 Why is the Dakar Niño/Niña variability amplified?

The simulations of the high-resolution regionally coupled model, ROM have shown that the Dakar Niño/Niña variability from March to April will intensify under global warming, particularly in the Dakar Niñas events. According to Oettli et al. (2016), the Dakar Niño is associated with changes in alongshore local surface wind and as shown Fig. 5, SST variability is well correlated with coastal winds, consistent with the findings by Oettli et al. (2016). To further understand this relationship, the surface wind changes are investigated in more detail here.

The standard deviation of meridional wind stress anomalies is presented in Fig. 8. In the observation, the high variability associated with the Azores high-pressure system (e.g., Davis et al., 1997) is found between 24°N and 30°N (Fig. 8a). Additionally, the meridional wind variability is relatively strong along the northwestern African coast down to 9°N as well. The ROM$_P$ simulation is able to effectively capture the spatial pattern of meridional wind variability, with the largest coastal variability centered around 20°N (Fig. 8b). However, ROMP somewhat overestimates the variability around 12°N and 20°W, resulting in forming two cores of high variability (in the observation, the second core around 12°N is much smaller and it is located more offshore as shown in Fig. 8a). Under global warming (Fig. 8c), the coastal wind variability is increased while the positions of the two cores remain  unchanged. In contrast, the meridional wind variability over the open ocean between 24°N and 30°N seems to decrease (Figs. 8b and c) indicating that the higher wind variability in the future might be more influenced by local effects around the coastal region.

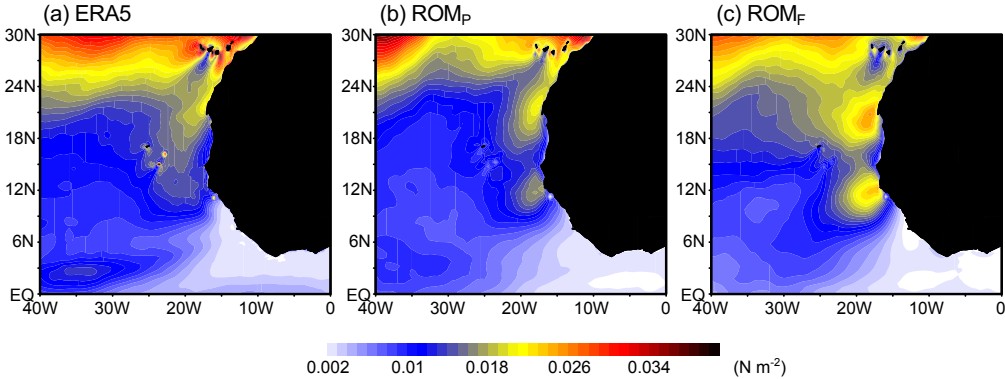

**Figure 8:** Standard deviation of the meridional wind stress in March for (a) ERA5, (b) ROM$_P$, and (c) ROM$_F$.

A possible explanation for the localized change in the surface wind is the land-sea heat contrast proposed by Bakun, (1990). According to Bakun (1990), in the context of global warming, terrestrial regions will heat up more intensely than the ocean, which will increase the land-sea heat contrast and consequently strengthen the equatorward coastal low-level jet and corresponding upwelling. Figure 9 shows the composite anomalies of 2m temperature during Dakar Niños and Dakar Niñas in ERA5 and ROM simulations. In ERA5, the 2m temperature anomalies reveal a land-sea thermal contrast, but the 2m

temperature anomalies over the land (signs are opposite to the Dakar Niños/Niñas near the coast) are located farther inland from the western African coast (around 0° to 20°E, Figs. 9a and b). ROM$_P$ can reproduce the terrestrial 2m temperature anomalies realistically in the case of Dakar Niño and Niña although its amplitude is weaker compared ERA5 (Figs. 9c and d). Conversely, the land-sea thermal contrast associated with the 2m temperature anomalies becomes more pronounced in ROM$_F$ (Figs. 9c fand f). During the Dakar Niño events, the magnitude of the cool anomaly over the continent is almost identical in 280 both present and future climate, but spatially, the land-surface temperature anomaly shifts toward the west, potentially weakening the zonal surface temperature gradient, particularly around the coastal region between 9°N and 12°N (Fig. 9b). In the case of the Dakar Niñas, the land surface temperature anomaly also shifts toward the west similarly to the case of the Dakar Niñas, but with a larger amplitude in ROM$_F$ than in ROM$_P$ (Figs. 9c and d). This situation can strengthen the zonal thermal contrast and the alongshore (upwelling-favorable) winds can be more effectively generated. In terms of climatology, the ROM 285 simulations show that the desert amplification (e.g., Cook and Vizy, 2015; Zhou, 2016) becomes more pronounced in the western Africa under RCP8.5 scenario (not shown).

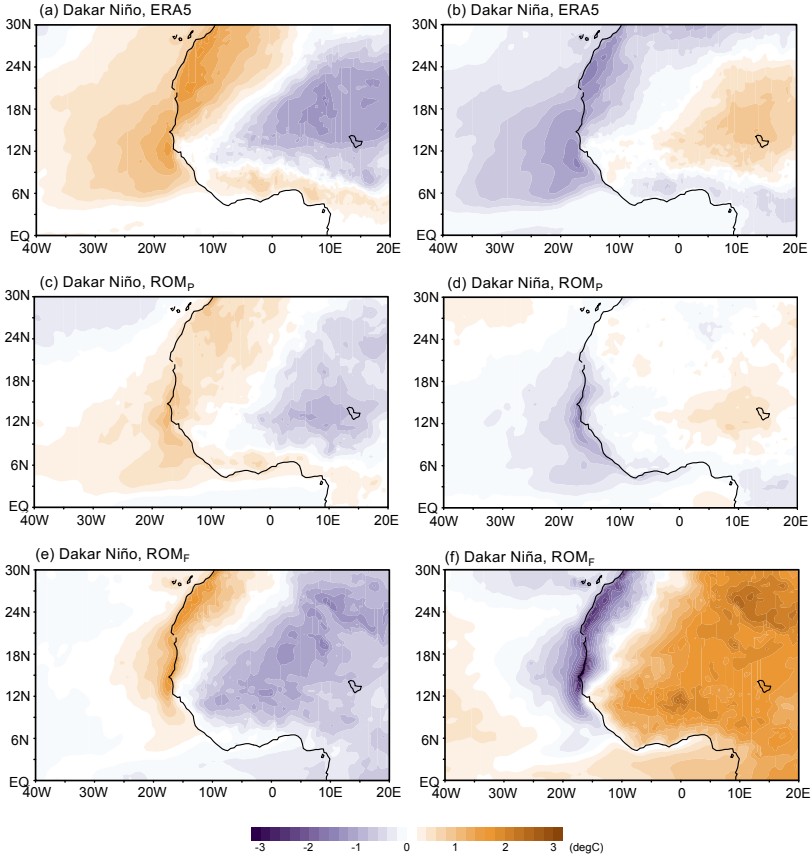

**Figure 9:** Composite anomalies of the 2m temperature averaged during (left) Dakar Niño/ /(right) Niña events in (top)ERA5, (middle) ROM$_P$
, and (bottom) ROM$_F$ in March.

This land-sea thermal contrast anomalies can also be seen by sea level pressure (SLP) anomalies (Fig.10). In ERA5, the SLP anomalies show a dipole pattern roughly over the Atlantic Ocean and the continent (Figs. 10a and b). While the SLP anomaly over the Atlantic is likely associated with the Azores high pressure, the SLP anomalies over the Sahara connect to the SLP anomalies over the Mediterranean and the SLP anomaly over the continent appears to be more responsible for creating the SLP zonal gradient along the coast, in particular, case of Dakar Niña (Fig. 10b). $ROM_P$ can represent this

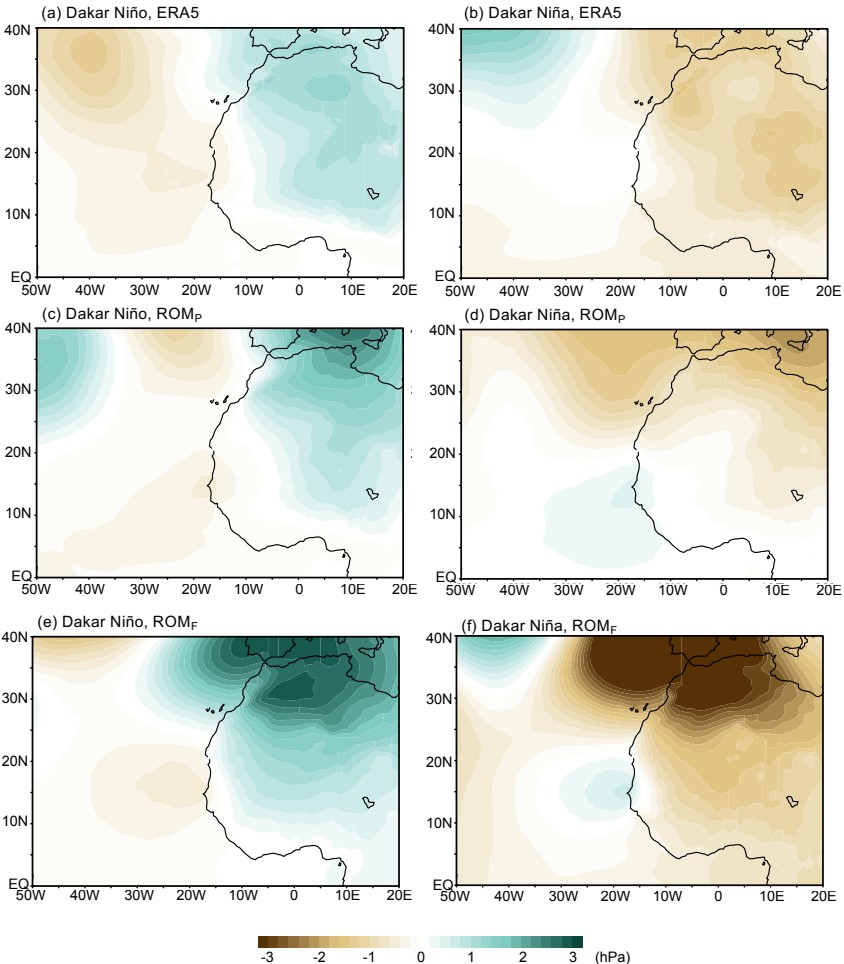

**Figure 10:** Same as Fig. 9 , but for sea level pressure in a wider domain.

SLP anomaly pattern that connects to the Mediterranean although the Azores high anomalies are not as clear as ERA5 (Figs. 10c and d). However, the cores of the continental SLP anomalies are located around 0 to 20°E, which is in line with ERA5. In ROM$_F$, the continental SLP anomalies intensify as the 2m temperature anomalies are strengthened (Figs. 9e, f, 10e and f). The SLP anomaly gradient runs across the coastal region of western African and this situation is favourable for meridional surface wind anomaly for Dakar Niño (reducing equatorward wind anomaly) and Nina (increasing equatorward wind anomaly). Notably, the Mediterranean SLP anomalies are intensively strengthened in both cases, leading to stronger Sahara SLP anomalies and creating a sharper zonal SLP gradient along the western Africa coast. This finding is consistent with reports that the inter-annual variability in the temperature is expected to increase in the future intensification in the Mediterranean region under global warming (Giorgi and Lionello, 2008) and that the Mediterranean SLP anomalies are also expected to be amplified in the future.

## 4.2 Heat Budget Analysis in Ocean Mixed Layer

As suggested by Figs. 6, 7, 9, and 10, the meridional surface wind variability is strengthened, which consequently can influence the ocean dynamics and thermodynamics including vertical motion, ocean currents, and surface turbulent fluxes. According to Oettli et al. (2016), surface heat flux plays a crucial role in generating Dakar Niño and Niña events and entrainment is secondary effect. Here we investigate which process will change in a future climate scenario. thereby affecting the Dakar Niño and Niña events in the future climate. To quantify this, here we examine the heat budget in the ocean mixed layer during Dakar Niño and Niña events. Following Vijith et al. (2020), the heat budget in the ocean mixed layer is estimated as follows,

$$\frac{\partial \text{SST}}{\partial t} = \langle -u\frac{\partial T}{\partial x}\rangle + \langle -v\frac{\partial T}{\partial y}\rangle - w_{OML}\frac{\Delta T}{D} + \frac{Q}{\rho C_p D} + R,$$

Here, the bracket indicates a quantity averaged within the ocean mixed layer. $D$ represents the ocean mixed layer depth (an output of ROM). $W_{OML}$ and $\Delta T$ denote the vertical velocity at the bottom of the ocean mixed layer and the temperature difference between in the ocean mixed layer and just below the ocean mixed layer (assuming the temperature within the ocean mixed layer is homogeneous vertically). $Q$ is the net surface heat flux. $\rho$ and $C_p$ are constant values of density (1000 kg m$^{-3}$) and specific heat of sea water (4200 J kg$^{-1}$ K$^{-1}$). $R$ is a residual term (the entrainment is included in this term) that we do not examine in this study. Note that the heat budget terms are estimated from monthly-mean data of velocity and temperature due to the limited data availability and therefore, some non-linear and transient components are missing in the heat budget.

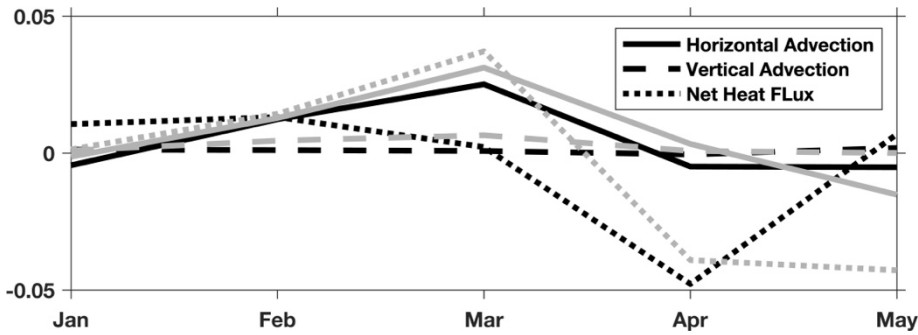

**Figure 11:** Monthly time series of lag-composite difference of (solid) horziontal advection, (dotted) surface net heat flux, and (dashed) vertical thermal advection between Dakar Niño and Dakar Niña events (Niño minus Niña) in (black) ROM$_P$ and (grey) ROM$_F$ within the Dakar Index box. March is lag=0. The unit is K day$^{-1}$.

In the present climate, the contributions of surface heat flux and horizontal thermal advection to the Dakar events is roughly comparable between January and March while the timing of peak differs (Fig. 11). Note that Fig. 11 shows the lag-composite difference between Dakar Niño and Niña to emphasize the anomalies during Dakar Niño (during Dakar Niña the anomalies should be opposite). Contrastingly, the vertical advection does not play a significant role in inducing Dakar events. The relatively large contribution of horizontal advection differs from the argument of Oettli et al. (2016). In agreement with Oettli et al. (2016), the mixed-layer heat budget of the ORAS5 reanalysis shows a crucial role of surface net heat flux anomalies

to Dakar Niño (the definition of the events in ORAS5 is same as ERA5 as shown in Fig. S6a) and a comparable magnitude of horizontal advection (note that vertical velocity data is not provided in ORAS5 monthly data) to ROM$_P$.

Each component of the heat budget increases in the future climate supporting our results of amplified Dakar Niño and Niña events. Specifically, the surface heat flux plays a more significant role in amplifying the Dakar Niño, in particular, in March. The horizontal and vertical thermal advection anomalies also intensify, with their magnitude of enhancement in the

345 future being almost identical between February and March (the differences between the current and future climate are 0.006K/day and 0.0057 K/day in March, respectively). For the vertical advection, the stronger stratification at the upper layer (Fig.7e) might enhance the contribution of vertical advection, in particular, in Dakar Niña (Fig. 7d). In the future, the climatology of surface ocean current is slightly weakened around our focus area (comparing ROM$_P$ and ROM$_F$ in the red rectangular in Fig. S5). However, the composite anomaly between Dakar Niño and Niña shows larger difference in the future

climate in the Dakar Index box (Fig. S5). This indicates that the stronger meridional wind variability along the coast can induce more local/regional surface ocean current change in the future than in the present climate. As Oettli et al. (2016) suggested, the ocean mixed layer depth tends to be thinner/thicker during Dakar Niño/Niña events (see Fig. S7). This might help to increase the contribution of surface heat flux in the future climate because the thinner mixed layer can reach more warming by stronger surface heat flux during Dakar Niño. Because of this amplified mechanism in March, the SST anomalies can be more

persisting until April in the future climate than in the present climate (Fig. 5).

The surface heat flux anomalies are divided into four elements as shown in Fig. 12. In the present climate, latent heat flux dominates from January to February, with shortwave radiation also playing a role in February. This distribution is similar

to Oettli et al. (2016). However, the contribution of shortwave radiation is not as dominant as Oettli et al. (2016). According to Oettli et al. (2016), shortwave radiation is a primary contributor to the heat flux anomaly for Dakar Niños. Our ROM simulations might underestimate the anomalous shortwave radiation, potentially due to climatological dust forcing (Pietikäinen et al., 2012). Chen et al. (2021) suggested that the Saharan dust influences significantly shortwave radiation flux directly and surface turbulent fluxes indirectly. The sensible heat and longwave radiation are quite minor, consistent with the findings of iOettli et al. (2016).

In the future, each component of surface heat flux becomes stronger, especially, latent heat flux and shortwave radiation. The enhancement of surface heat flux in March is attributed mainly to the latent heat flux and secondarily to the shortwave radiation. The enhancement of latent heat flux can be explained by the stronger alongshore wind variability as shown in Fig. 8. It is necessary, but out of scope to investigate how the Saharan dust anomaly and cloud anomaly affects the surface heat flux and correspondingly Dakar Niño in the current study. This will be addressed in the future works.

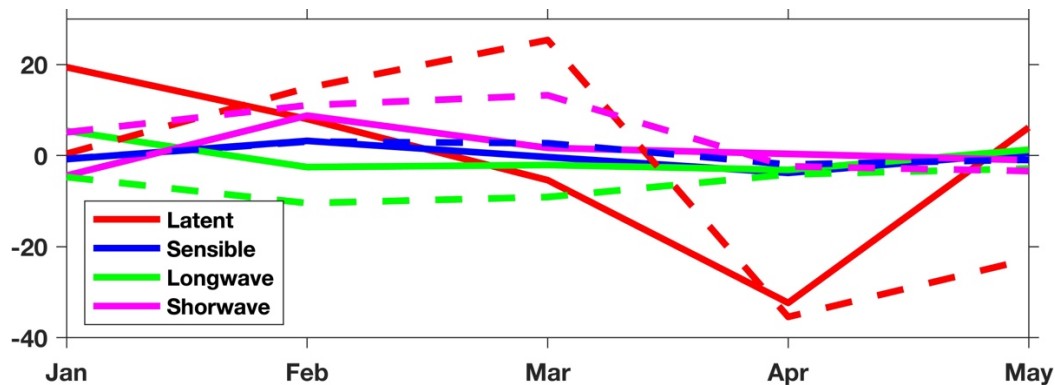

**Figure 12:** Monthly time series of lag-composite difference of latent heat flux (red), sensible heat flux (blue), longwave radiation (green), and shortwave radiation (magenta) for $ROM_P$ (solid) and $ROM_F$ (dashed) with the Dakar Index box. The unit is W/m$^2$.

### 4.3 Conclusion and future works

This study has investigated the future change of Dakar Niño variability in March employing the high-resolution regionally-coupled model, ROM, comparing the 1980-2010 and 2066-2099 periods under the highest emission scenario. Our model simulations show an intensification of interannual variability in the SST along the northeastern tropical Atlantic with a notable increase in the amplitude of Dakar Niña (cool SST anomaly) events. This result is consistent with Yang et al. (2021) while they focus on the basin-scale variability in the north tropical Atlantic. In contrast, Prigent et al. (2023) reported a weakening of the Benguela Niño under global warming. This result is in contrast to our results, underscoring the need of demonstrating insightful comparison between these two coastal climate modes to discuss similarity and dissimilarity of Dakar and Benguela Niños. For example, recently Chang et al. (2023) showed the different response of eastern coastal upwelling

systems to climate change in northern and southern hemispheres using a set of HighResMIP model, emphasizing the importance of such comparative studies.

The stronger variability of SST in the SMFZ under global warming can be explained by the stronger surface heat flux anomalies (mainly latent heat flux and secondarily shortwave radiation) partially associated with the local alongshore wind variability. The contribution of horizontal and vertical thermal advection anomalies also tends to amplify the Dakar Niño/Niña while their role is secondary. The alongshore wind variability can be enhanced by the well-developed thermal contrast anomaly around the west African coast as discussed by Bakun (1990). Moreover, we found that the corresponding Saharan sea level pressure (SLP) anomalies are extended from the Mediterranean region and the Mediterranean SLP is strengthened. In addition, the stronger ocean stratification at 40m depth might also cause the reinforcement of the Dakar Niño/Niña variability. This stronger stratification is due to the vertically-heterogeneous warming between the surface and subsurface (e.g., Vazquez et al., 2023). The ocean surface current anomaly during Dakar Niño and Niña can be also changed by the stronger meridional wind stress in the future. Especially, the ocean current anomaly changes in the Dakar Index box.

Our discussion and argument focus on local-/regional-scale changes of surface wind and land-sea thermal/surface pressure contrast. However, it is essential to acknowledge broader climate teleconnections. As the previous studies suggest, the tropical Pacific inter-annual variability like El Niños tends to initialize the north tropical Atlantic variability including Dakar Niños via atmospheric bridge (e.g., Oettli et al., 2016; Lopez-Parages et al., 2020), we will need to consider such teleconnection and its future change. In addition, over the north Atlantic, other dominant climate mode such as North Atlantic Oscillation (NAO; e.g., Hurrell et al., 2001) plays a crucial role in climate and weather variability over the Euro-Mediterranean region modulating the Azores high-pressure system (e.g., Brandimarte et al., 2011; Lopez-Moreno et al., 2011). Therefore, it will be necessary to explore the linkages between Dakar Niño and other climate modes such as NAO and ENSO in order to reach a more comprehensive understanding of how these patterns interact and evolve under global warming.

*Code Availability*

The codes used in this study can be accessed at Zonode repository, 10.5281/zenodo.10244333.

*Data Availability*

The data used in this study can be accessed at Zonode repository, 10.5281/zenodo.10244333.

*Author Contributions*

SK, VR, and WC contributed to conceptualizing the study and had discussions on the results. SK mainly demonstrated the analyses. DSV and WC have performed all the simulations used in this study. CG and MLB contributed to improving interpretation of the results and discussion. All co-authors contributed to drafting the manuscript and revising it.

*Competing Interests*

All co-authors declare that there is no conflict and competing interests.

*Acknowledgement*

All authors would like to express their gratitude to two anonymous reviewers for their critical, helpful and constructive comments and discussion. Thanks to those, our manuscript could be improved and the discussion on the results be significantly insightful. SK was supported by EU Horizon 2020 TRIATLAS project (agreement number: 817578) and Giner de los Ríos

2021/22 program by University of Alcalá. RV was supported through a doctoral grant at University of Ferrara and University of Cadíz, the Spanish Ministry of Science, Innovation and Universities (I+D+I PID2021-128656OB-100) and the Plan Propio UCA 2022-23. WC and CG were funded by the Alcalá University project (PIUAH21/CC-058) and the Spanish Ministry of Science, Innovation and Universities, through grant (I+D+I PID2021-128656OB-100). DVS was supported by the Germany-Sino Joint Project (ACE, No. 2019YFE0125000 and 01LP2004A) and MHESRF scientific task № FMWE-2024-0028. MLB

has received funding from the European Union's Horizon 2020 Research and Innovation Program for the project BENGUP under the Marie Skłodowska-Curie grant agreement ID 101025655. The simulations were performed at the German Climate Computing Center (DKRZ), granted by its Scientific Steering Committee (WLA) under project ID ba0987. All authors would like to express their gratefulness to Prof. Noel S. Keenlyside at the University of Bergen/BCCR for his constructive comments and discussions on this work.

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
