# Peer review of "Dakar Niño under global warming investigated by a high-resolution regionally coupled model"

_EGUsphere, 2023_

## Author Comment (AC1)

**Review for "Dakar Niño variability under global warming investigated by a high resolution regionally coupled model" by Koseki et al.**

The study by Koseki et al. investigates how sea surface temperature (SST) variability at the Senegalese-Mauritania coast associated with so called Dakar Niños might change in the future. Utilizing a regionally high-resolution coupled climate model they find that the Dakar Niño variability increases under the RCP8.5 scenario. This is explained by an increase in the wind variability and higher ocean stratification under global warming.

The manuscript is mostly well structured and written and provides interesting results on the future of SST variability in the Northeastern Tropical Atlantic. However, I find that it is sometimes hard to follow the presentation of the results and that the investigation of the processes needs some work. Also, there are some recent studies dealing with similar questions that should be taken into account. I summarize my major comments and list minor points and specific comments below.

**REPLY:** We gratefully appreciate the reviewer for the constructive comments. Following the comments, we have corrected our manuscript by adding more analysis and reply point-by-point as follows. Please note that the tracked changes in the revised manuscript are shown in blue color and the number of lines and figures are for the revised manuscript.

**Major comments:**

**A) Definition of Dakar Nino**

It is not really clear to me what exactly this study regards as Dakar Niño variability. Is it all interannual SST variability in the region considered here (i.e. from 7ºN to 21ºN)? Everything related to the position of the front (line 37/38)? Or just SST anomalies occurring in the Dakar Niño Index box which is much more confined? Is the peak in SSTA variability around 20ºN in boreal summer also related to Dakar Niños? Please provide a definition and make sure to be consistent throughout the manuscript. Also, please indicate the Dakar Niño box in Figure 4.

**REPLY:** Thank you very much for the helpful comment. Throughout the manuscript, we focus on SST interannual variability as the box-mean (9N-14N and 20W-17W defined by Oettli et al., 2016), which is defined as Dakar Index in Oettli et al. (2016). The events of Dakar Niño/Niña in **March** are defined as the Dakar Index plus/minus ± the standard deviation. According to Oettli et al. (2016) who found this event first, March variability is maximum and we follow their definition. While we described the definition, the location of description was not adequate. Therefore, we moved the

definition of Dakar Niño/Niña events in this study to Line 171-173 in the revised manuscript to improve the readability.

In addition, as another reviewer suggests, we added a time series of Dakar Index and events of Dakar Niño and Niña in Fig.R1 as follows, as well as its description in the revised manuscript. Please see lines 171-177. Due to this new figure, we added a box of the Dakar Index in Fig.3.

[Figure]

Fig.R1. Time series of Dakar Index (detrended SST averaged 9°N-14°N and 20°W-17°W) for (top) ERA5, (middle ROM$_P$, and (bottom) ROM$_F$ The orange and blue dots indicate Dakar Niño and Niña events defined in this study, respectively.

**B) Processes behind Dakar Niño variability**

**I believe that more analysis is needed on the processes by which Dakar Niños and Niñas are driven in the simulations and how they change in the future. For example, Oettli et al. (2016) argue that heat fluxes are important for the generation of Dakar Niños but they are not considered here. The same goes for changes in the depth of the mixed layer.**

**REPLY:** Thank you very much for the helpful comment. Indeed, the process of the changes in Dakar Niño/Niña events should be discussed. Here, we estimated the more important components of the ocean mixing layer heat budget (according to Oettli et al., 2016) and Figure R2 shows the composite differences between Dakar Niño and Dakar Niña in current (1980-2010) and future climate (2069-2099). Please note that we examine surface net heat flux (proposed by Oettli et al., 2016) and vertical advection. Our analyses indicate that the vertical motion variability intensifies and is deepened due to reinforced meridional wind-stress variability under global warming (Figs. 6 and 8, please note that the numbering of figures have been changed) and therefore, here, as a first order, we compare surface heat flux and vertical thermal advection roles. The vertical thermal advection (Vadv) is defined as in Vijith et al. (2020)

$$\text{Vadv} = -w_{oml}\frac{\Delta T}{D}$$

here, $w_{oml}$ is the vertical velocity (m/s) at the bottom of ocean mixing layer (m), $D$ is the ocean mixing layer depth (model's output), and $\Delta T$ is the difference between the ocean mixing layer temperature and the temperature at the layer just below the ocean mixing layer.

Please note that the vertical thermal advection is estimated using monthly-mean data because of the data availability. Therefore, the transient component of vertical advection is not included, which could lead to some under/overestimation.

As Fig.R2, our analysis shows that in the current climate, surface heat flux is relatively more responsible for warming SST, but vertical advection also explains the warming SST in the Dakar box. On the other hand, in the future climate, the role of vertical advection is extensively increased. This result supports our frist argument: stronger meridional wind variability can excite the vertical motion variability and consequently, Dakar Niño/Niña events can be reinforced. In addtion, as Fig. 7e shows, the upper ocean layer become much warmer than sub-surface ocean layer between 40m depth. As the reviewer mentions as below, the ocean is more stratified in the future climate. This indicates that the contribution of vertical thermal advection could increase since $\partial T/\partial z$ increased in the upper layer. We added this discussion and Figure as Fig.11 in the Section 4. Please see lines 294-320. Moreover, we rephrased the last part of Abstract referring to this result. Please See line 21-23.

[Figure]

Fig.R2 Monthly time series of lag-composite difference of (solid) surface net heat flux and (dashed) vertical thermal advection between Dakar Niño and Dakar Niña events (Niño minus Niña) in (black) 1980-2010 and (grey) 2069-2099. March is lag=0. The unit is K day$^{-1}$.

Also, we analyzed the composite difference of ocean mixing layer in the Dakar Index box between Dakar Niño and Dakar Niña as shown in Fig.R3.

[Figure]

Fig.R3 Monthly time series of lag-composite difference of ocean mixing layer depth between Dakar Niño and Dakar Niña events (Niño minus Niña) in (black) 1980-2010 and (grey) 2069-2099. March is lag=0. The unit is m.

As Oettli et al. (2016) shows, the mixing layer depth tends to be shollower during the Dakar Niño in our simulation. Interestingly, the composite difference in the ocean mixing layer is larger in the future climate, especially from February to April. One of the possible reasons for this could be the meridional wind variability and the, the mechanical contribution to deepening the mixng layer can increase the difference in ocean mixing layer. However, the ocean mixing layer is also function of temperature and salinity, which could be difficult to make an attribution. We added this figure in Supplementary Information as Fig.S5 and some description. Please see lines 294-320.

It is also not clear to me why stronger stratification should lead to higher variability. Stratification is also increasing in the equatorial Atlantic but there, the argument is that this hinders subsurface-surface coupling and thus leads to weaker variability.

**REPLY:** Thank you very much for giving an oppostunity to discuss our argument more. For this statement, we show the composite of $\Delta T$ in the vertical thermal advection in March during Dakar Niña events for $ROM_P$ and $ROM_F$ in Fig.R4.

[Figure]

Fig.R4. The composite of $\Delta T$ during Dakar Niña in March for (left) $ROM_P$ and (right) $ROM_F$.

In both cases, there is a strong vertical temperature gradient along the coast, $ROM_F$ shows relatively larger $\Delta T$. In addition, offshore areas also have larger $\Delta T$ entirely. Given the same vertical velocity anomaly emerges in present and future climate, the future climate will have more vertical thermal advection during Dakar Niña events.

On the other hand, during Dakar Niño events, there is not difference in the mixing layer depth (already ckecked). Therefore, under global wamring, espeacially, Dakar Niña will be more reinforced as shown by our Fig. 7.

It is further argued that correlation between the DNI and ocean temperatures at greater depth means stronger SST variability (line 189). Why would that be?

**REPLY:** We agree that this argument might be too speculative. The deeper connection between DNI and ocean temperature just gurantees that the Dakar Niño/Niña events would be deeper in the future, not larger variability. Therefore, we removed this statement and instead, we added "indicating that Dakar Niño and Niña events would influence deeper ocean in the future" . Please see lines 207-208.

In the conclusions "stronger vertical velocity variability" (279) is mentioned. Where is this shown?

**REPLY:** As in the previous case, this argument might also be too speculative due to less process-wise analysis. As the reviewer suggests and we have done in Fig.R2, which suggest that the role of vertical thermal advection is responsible for enhancing Dakar Niño and Niña events in the future. Therefore, we rephrased here as "inducing a stronger vertical thermal advection". Please see lines at 333.

**C) Literature to be taken into account**

**There are some recent studies dealing with the future of the variability in the Dakar Niño region that should be cited here:**

**Yang et al. (2021) find increasing variability in the Northeastern Tropical Atlantic under global warming. Chang et al. (2023) look at changes in temperature and wind patterns in eastern boundary upwelling regions in HighResMip simulations. Also, a study on the future of Benguela Niños was recently published (Prigent et al., 2023) and could be mentioned in line 142.**

**REPLY:** Thank you very much for providing information. We added these articles in discussion of the revised manuscript and corresponding descriptions. Please see lines 326-331.

**Minor points:**

**1) Model validation**

**While I like the comparisons shown in Fig. 2, they could be improved and further complemented with line plots. The axis labels are very small and the color scale of (e) to (h) is hard to interpret (this also applies to Fig. 7). I would also find it more instructive to show SST in °C. In addition, one could show a line plot with SST and its standard deviation averaged over certain latitude bands overlaid for the different data sets and model simulations. This would facilitate an easier comparison in terms of timing and amplitude. I believe that it would probably show that the season of highest variability in the model is shifted towards later in the year (April instead of March) which makes the focus on March later in the analysis a bit questionable.**

**REPLY:** Thank you very much for constructive suggestions. We improved the figure following the comments. Regading the latitude band plot, we added the figure as Supplementary Information new Fig.S1. Regarding the unit of SST, we added a new analysis of surface heat flux and vertical thermal advection as above, and the unit of temperature in this computation is K. Therefore, it could be reasonable to have the unit of K.

Yes, our model has a peak of variability by about one month later. However, to compare with the observation, it could be better to invesigate the same month between models and observations. Our model still shows a strong variability in March. We added this justification in the revised manuscipt. Please see line 146-150.

**Further, in line 120, it is stated that the model has a warm bias. Where? What is the general pattern of SST difference to observations?**

**REPLY:** This statement is about ERA5 bias compared to the ESA satellite data, not our model. As we cite, Vázquez et al. (2022) have done more model assessment with the same models and same experiments. Additionally, you can find more details this fact in Vázquez et al. (2023). Therefore, this study has not performed detailed assessment of the model performance.

**2) Seasonality of SST in the region**

**The seasonal cycle in SST shown in Figure 2 is mainly explained in terms of a meridional movement of the front driven by meridional currents. However, I believe that it is largely impacted by the upwelling season in the southern part of the region, i.e. cooler waters are upwelled to the surface in February to April.**

**REPLY:** Thank you very much for this comment. Actually, we had already mentioned the role of upwlling at line 105-107 citing some references.

**Regarding the seasonal migration of the Mauritania current (line 105), is there a reference for this? Does it fit with the findings by Klenz et al. (2018)?**

**REPLY:** We cited Klenz et al. (2018) as a reference of Mauritania Current. Plesse see lines 110-111.

**3) Role of remote forcing**

**It is stated (lines 42 to 44, also 116 to 118) that Benguela Niño events are stronger than Dakar Niños because of the additional role of remote forcing from the Equator. Is this something that has actually been shown (in this case, please provide a reference) or just an assumption of the authors?**

**REPLY:** The remote foricng for Benguela Niño has been a well-established mechanism and therefore, we have already cited related papers. At lines 42-44, the references were missed, then we added the literatures. As far as we know, there is no study on direct comparison between Benguela and Dakar Niño. Therefore, the statement at lines 116-118 might be speculative and we deleted the sentence "and the Benguela Niño/Niña has more larger intensity than Dakar Niño/Niña because of the strong remote influence from the equator via equatorial and coastal Kelvin waves".

**4) Correlation with Dakar Niño index (Figure 4)**

**How do the correlations in Figure 4 change when they are based on April instead (see comment 1 above)? Also, I am not sure the propagating signal described is really propagating (associated with a Rossby wave). Couldn't the SST anomaly just decrease first at the coast due to coastal processes and linger around for longer offshore?**

**REPLY:** As we replied above, we focus on March in order to keep the consistency with the observation. We would expect that there might be some differences, but some of characteristics might be quite similar.

Regarding the Rossby wave, our statement here just indicate a possibility of Rossby wave as Martín-Rey and Lazar (2019) showed. Their arguement is the connection between boreal spring variability, Atlantic Meridional Mode (AMM), and summer Atlantic Zonal Mode (AZM). AMM signal can propagate as Rossby Wave, and it reflects in the western boundary as Kelvin wave propagating eastward and genetating AZM. Our lag correlation map might give an indication of AMM after Dakar Niño, but it is unknown the connection between Dakar Niño and AMM and this will be one of possible futute works, which is out of scope of this study.

**Specific comments:**

**- At several instances, "reinforced" is used when probably "strengthened" or "increased" is meant (e.g. line 23, 206, 217, 226, 269,…)**

**REPLY:** Changed.

**- line 17/18 and 186 to 191: Here, it is not clear whether the correlation gets stronger or whether high correlation is extending deeper in the ocean. Please rephrase to make this clear.**

**REPLY:** At the upper ocean, the correlation is relatively stronger and the correlation is much deeper. As we replied above, we rephrase this part. Please see lines 207-208.

**- line 33: "feature" instead of "exhibit"**

**REPLY:** Changed.

**- line 41: "driven by"**

**REPLY:** Added.

**- line 52: In contrast to what?**

**REPLY:** Chnaged to "However".

**- line 53: "even in" instead of "until"**

**REPLY:** Changed.

**- line 58: "recently" instead of "timely"**

**REPLY:** Changed

**- Figure 1: What is shown by the shading?**

**REPLY:** That is the topography height in NEMO. We added it in the caption.

**- line 105: What is "inversely" referring to? When are the trade winds relaxing?**

**REPLY:** Yes.

**- line 111: "second maximum" in terms of timing or location?**

**REPLY:** Here, we mean both SST gradient and interannual varability peaks are almost overlapping as shown in Fig.2. We added "at the same timing and location".

**- line 116: "has larger" (without "more")**

**REPLY:** The sentence was removed replying to the comment above.

**- line 128: "focus region"**

**REPLY:** Changed.

**- line 138/139 "K" or "°C" instead of "degree"**

**REPLY:** Changed.

**- line 149: ORA-S5 is a reanalysis product.**

**REPLY:** Changed.

**- line 152: Please rephrase this sentence.**

**REPLY:** Rephrased. Please lines 158-159.

**- caption to Fig. 4: "January" (typo)**

**REPLY:** Corrected.

**- line 173: "south of" instead of "below"**

**REPLY:** Changed.

**- line 177: What does "somewhat simulated well" mean? Please rephrase.**

**REPLY:** Chnaged to "to some extent "and changed well to "realistically".

**- line 178: "is correlated", "SST anomalies develop"**

**REPLY:** Corrected.

**- line 179: "compared to ERA5"**

**REPLY:** Correcnted.

**- line 181: Should be "S2"**

**REPLY:** Changed, but please note that this I snow Fig.S3 because of an addition new Supplementary figure.

**- line 199: "1 standard deviation"**

**REPLY:** Changed, but please note that the senence has been moved to lines 171-173 as we replied above.

**- line 284: "is" instead of "can be also"**

**REPLY:** Corrected.

**References:**

Chang, P., G. Xu, J. Kurian et al., 2023: Uncertain future of sustainable fisheries environment in eastern boundary upwelling zones under climate change, Comm. Earth & Environment, https://doi.org/10.1038/s43247-023-00681-0

Klenz, T., Dengler, M., and Brandt, P., 2018: Seasonal variability of the Mauritania Current and hydrography at 18°N. Journal of Geophysical Research: Oceans,123, 8122–8137. https://doi.org/10.1029/2018JC014264

Prigent, A., Imbol Koungue, R., Lübbecke, J.F. et al. Future weakening of southeastern tropical Atlantic Ocean interannual sea surface temperature variability in a global climate model. Clim Dyn (2023). https://doi.org/10.1007/s00382-023-07007-y

**Yang, Y., L. Wu, Y. Guo et al., 2021: Greenhouse warming intensifies north tropical Atlantic climate variability, Sci. Adv. 2021;7:eabg9690**

**REPLY:**  Thank you very much for the information on further literature. We added these articles and descriptions referrign to them. Please see lines 326-331.

---

## Author Comment (AC2)

Review of "Dakar Niño variability under global warming investigated by a high-resolution regionally coupled model"

by Koseki, S., Vázquez, R., Cabos, W., Gutiérrez, C., Sein, D. V., and Bachèlery, M.-L.

General comments

The aim of this work is to characterize the future of the coastal upwelling regions off the Senegalese coast, under the IPCC' RCP8.5 pathway (i.e., highest emission scenario), using a high-resolution regionally-coupled model. This is an important topic, because the future of the upwelling regions around the world is still under investigation and subject to debates. In Benguela and California systems, a consensus for a positive trend in upwelling-favorable winds seems to exist (Sydeman et al., 2014), the signal is less clear for the Canary region. Particularly the response of marine ecosystems to climate change/global warming is still uncertain (e.g., Xiu et al., 2018). Also, the upwelling regions in eastern tropical oceans are known to be not well represented in climate models (Richter, 2015). So a better knowledge of their variability and underlying mechanisms is still needed.

However, the current study presents several issues, some major, some minor, that need to be addressed before publication. The detail of these issues is given below. I therefore recommend major revisions.

**REPLY:** We greatfully appreciate the reviewer for the constructive comments. Following the suggestions, we have corrected our manuscriprt by adding more analysis and reply point-by-point as follows. Please note that the tracked changes in the revised maniscript are shown in blue color and the number of lines and figures are for the revised manuscript.

Major comments

In this section, the major issues are detailed.

Dakar Niño

This coastal Niño phenomenon is, like the ENSO or other coastal Niños, is a recurring climate pattern, characterized by sea-surface temperatures (SST) warmer by a few degrees than normal, with only a few occurrences in a decade. This is different from the definition given at L.37-38.

**REPLY:** Thank you very much for this comment. We rephrased it for clarity as: "and some extreme events of SST warm anomalies are called Dakar Niño". Please see lines 38-39.

**This is also different from the Dakar Niño Index (DNI) which depicts the temporal variability of the SST, averaged over the 21°–17°W, 9°–14°N region. In this perspective, the title could be changed to reflect either the exploration of the variability of the DNI under RCP8.5 or the future of Dakar Niño under RCP8.5. This is particularly true regarding Figs. 5 and 6. The former is using the DNI, while the latter is focusing on Dakar Niño and Dakar Niña.**

**REPLY:** Thank you very much for this comment. This study first started analyzing Dakar Niño Index between present and future climate and our main discussions are based on Dakar Niño and Niña events. Therefore, we would prefer to keep "Dakar Niño" in the title. However, "variabiltiy" has been omited from the title.

In response to the comment below, we have added a new figure (Fig.4) showing DNI and frequency of Dakar Niño/Niña events in ERA5 and ROM simulations. This new figure might help to avoid confusion. Please see line 171-175.

**The existence of a Dakar Niño (or Niña) is the result of complex land-sea-atmosphere interaction system. According to Oettli et al. (2016), the coastal, alongshore, wind variability, the mixed-layer depth anomaly, and the modulation of the mixed-layer temperature (mostly due to the shortwave radiation variations), are the necessary components to develop a Dakar Niño. The current work is mainly focusing on the coastal winds and some atmospheric variables (sea-level pressure, 2m temperature,...), forgetting about the other components of the dynamical system. It would be preferable, in this work, to also discuss about the evolution of the costal ocean mixed-layer, and its heat budget, under the RCP8.5. At L.257, there is a mention of the land-sea thermal contrast anomalies. It would be interesting to discuss how the contrast helps to maintain, or not, the favorable alongshore winds.**

**REPLY:** Thank you very much for the helpful comment. Indeed, the process of the changes in Dakar Niño/Niña events should be discussed. Here, we estimated the more important components of the ocean mixing layer heat budget (according to Oettli et al., 2016) and Figure R2 shows the composite differences between Dakar Niño and Dakar Niña in current (1980-2010) and future climate (2069-2099). Please note that we examine surface net heat flux (proposed by Oettli et al., 2016) and vertical advection. Our analyses indicate that the vertical motion variability intensifies and is deepened due to reinforced meridional wind-stress variability under global warming (Figs. 6 and 8, please note that the numbering of figures have been changed) and therefore, here, as a first order, we compare surface heat flux and vertical thermal advection roles. The vertical thermal advection (Vadv) is defined as in Vijith et al. (2020)

$$-w_{oml}\frac{\Delta T}{D}$$

here, $w_{oml}$ is the vertical velocity (m/s) at the bottom of ocean mixing layer (m), $D$ is the ocean mixing layer depth (model's output), and $\Delta T$ is the difference between the ocean mixing layer temperature and the temperature at the layer just below the ocean mixing layer.

Please note that the vertical thermal advection is estimated by monthly-mean data because of the data availability. Therefore, the transient-compoent of vertical advection is not included here and there could be some under/overestimation.

As Fig.R2, our analysis shows that in the current climate, surface heat flux is relatively more responsible for warming SST, but vertical advection also explains the warming SST in the Dakar box. On the other hand, in the future climate, the role of vertical advection is extensively increased. This result supports our frist argument: stronger meridional wind variability can excite the vertical motion variability and consequently, Dakar Niño/Niña events can be reinforced. In addition, Fig. 6e shows that the upper ocean layer becomes much warmer than subsurface ocean layer between 40m depth. As the reviewer mention as below, the ocean is more stratified in the future climate. This indicates that the contribution of vertical thermal advection could increase since $\partial T/\partial z$ increased in the upper layer. We added this discussion and Figure as Fig.11 in the Section 4. Please see lines 294-320. Moreover, we rephrased the last part of Abstract referring to this result. Please see line 21-23.

[Figure]

Fig.R1 Monthly time series of lag-composite difference of (solid) surface net heat flux and (dashed) vertical thermal advection between Dakar Niño and Dakar Niña events (Niño minus Niña) in (black) 1980-2010 and (grey) 2069-2099. March is lag=0. The unit is K day[-1].

Also, we analyzed the composite difference of ocean mixing layer in the Dakar Index box between Dakar Niño and Dakar Niña as shown in Fig.R3.

[Figure]

Fig.R2 Monthly time series of lag-composite difference of ocean mixing layer depth between Dakar Niño and Dakar Niña events (Niño minus Niña) in (black) 1980-2010 and (grey) 2069-2099. March is lag=0. The unit is m.

As Oettil et al. (2016) shows, the mixing layer depth tends to be shollower during the Dakar Niño in our simulation. Interestingly, the composite difference in the ocean mixing layer is larger in the future climate, especially from February to April. One of the possible reasons for this is because the meridional wind variability increased and the mechanical contribution to deepening the mixing layer can increase the difference in ocean mixing layer. However, the ocean mixing layer is also a function of temperature and salinity, so it could be difficult to make an attribution. We added this figure in Supplementary Information as Fig.S4 and some descriptions. Please see lines 294-320.

Regarding the thermal contrast between land and sea, our argument is based on Bakun's hypothesis combining sea-level pressure (SLP) anomalies in Fig 10. In both cases of Dakar Niño and Niña, continental SLP anomalies show steeper zonal gradient running along the coastal region. This gradient can be favorable for meridional wind anomalies for Dakar Niño (reducing equatorward anomaly) and Niña (increasing equatorward wind anomaly), which is consistent with Bakun's hypothesis. We added this discussion in the revised manuscript. Please see lines 287-288.

**A time series of the DNI is also needed for the periods 1980-2010 and 2069-2099, to provide the reader with the number and intensity of the events. This is also particularly important to understand Fig. 4.**

REPLY: Thank you very much for the helpful comment. We plotted the time series of the Dakar Index and events of Dakar Niño and Niña (Fig.R3 as follows). From this plot, the frequency of Dakar Niño and Niña is 7/9 and 8/6 in the present and future climate,

respectively. It seems that there would be not a large change of the event frequency with more events being over 1 K. We added this description in the revised manuscript. Please see lines 171-175.

[Figure]

Fig.R3. Time series of Dakar Index (detrended SST averaged 9°N-14°N and 20°W-17°W) for (top) ERA5, (middle) $ROM_P$, and (bottom) $ROM_F$ The orange and blue dots indicate Dakar Niño and Niña events defined in this study, respectively.

**Target of the study**

**There is some confusions throughout the main text. The title indicates that this work is focusing on the Dakar Niño, which is a coastal, phenomenon with specific characteristics (see previous sub-section), but the Senegalese–Mauritania Frontal Zone (SMFZ) is also often referred, introducing some sort equivalence in the reader's mind, between two different phenomena.**

**REPLY:** Thank you very much for pointing this out. As Fig. 2 shows, the high interannual variability of SST is found around the SMFZ and some extreme events are referred to Dakar Niño/Niña. Therefore, while the SMFZ is the name of an area with

certain climatic features, Dakar Niño/Niña are the phenomena. To make this point clearer, we modified a sentence in the introduction. Please see lines 38-39.

**Also, this study discusses the intensity and the location changes of the Dakar Niño between present and future periods, but doesn't tackle the frequency change. It would be informative to also present if, under RCP8.5, the number of Dakar Niño events is likely to increase/decrease.**

**REPLY:** As we replied above, we added the new plot of Dakar Niño Index and the number of events in Fig. 4. Accoridng to this plot, there are no large change in the frequency of Dakar Niño and Niña events between the present and futuree climate

**The Bakun hypothesis (or Upwelling intensification hypothesis, Cropper et al., 2014) is introduced at L.235. This is an important topic to discuss when it comes to the future of tropical upwelling regions, because several studies have been dedicated to corroborate or contradict the Bakun hypothesis. See for example Oettli et al. (2021, p.255–256) for a discussion on this. I would recommend to better highlight how this study seems to corroborate the upwelling intensification hypothesis.**

**REPLY:** As we replied to the comment above, our result is supporting the Bakun's hypothesis during Dakar Niño and Niña events: In particular, SLP anomalies over the land are favorable conditions for generating meridional wind anomalies. We added more discussion on this point. Please see lines 287-288.

On the other hand, it is still an open question why the SLP anomalies over the land surface are like that while the connection with the Mediterranean seems stronger in the future than in the present. It will be interesting to investigate this pattern as a future work.

**Clarity**

**The global structure of the text is not clear and needs to be rethought, because it is often difficult to understand what the authors are describing and putting emphasis on.**

**Throughout the text sometimes appears some vague statements which need to be clarified:**

**REPLY:** Thank you very much for helpful suggestions. We solved this vagueness of our description.

**L.64-68: The apparent opposite conclusion on model resolution between Sylla et al. (2022) and Vázquez et al. (2022) should be better emphasized and explained, because of its implication for the current work.**

**REPLY:** This might be caused by eddy-permitting model or not. Vázquez et al. (2022) utilized the eddy-permitting models around the Northeastern tropical Atlantic. This study also uses the model. We already added this point. Please see line 69.

**L.158: What is the meaning of this anomaly pattern for the Dakar Niño?**

**REPLY:** This pattern is the anomaly of climatological fields, therefore, it might be difficult to discuss the indication for Dakar Niño from this plot. For the climatological aspects, we already described some indications. Please see lines 171-174.

**L.188-189: "This deeper connection of ocean temperature in ROMF can be indicative of the stronger SST variability". This is unclear what the authors are saying here. This needs to be clarified.**

**REPLY:** This was too speculative and therefore, we changed this part as another reviewer also pointed out. Please see lines 207-208.

**L. 228: "[...] indicating that the wind variability might be more relevant due to local effects". This also is unclear. What are those local effects?**

**REPLY:** This statement is based on Fig.8: the wind variability over the subtropical Atlantic is not changed, but the coastal wind variability increases in the future. Therefore, the anticyclone variability is not responsible for the wind variability along the western African coast. To clarify which local effects are responsible for the higher wind variability, we analyzed the surface temperature and SLP anomalies. We modified the sentence. Please see lines 246-247.

**L.278-280: How? What would be the mechanism?**

**REPLY:** To clarify this speculation, we performed the heat budget between surface heat flux and vertical thermal advection. Please lines 294-320.

**L.283-285: Again, how? What would be the mechanism?**

**REPLY:** To clarify this speculation, we performed the heat budget between surface heat flux and vertical thermal advection. Please lines 294-320.

**Figs. 5 and 6: The mix between DNI and Niño/Niña events makes things difficult to follow.**

**REPLY:** To make it clearer, we added the new Fig.4 and corresponding descriptions. Please see lines 171-177.

**L.292-293: This statement is not clear. What are the other climate modes? Please clarify.**

**REPLY:** The other climate modes we argue here are NAO and ENSO. We added these modes. Please line 346.

**Specific comments**

**L.25: The region of the Senegalese–Mauritania Frontal Zone (SMFZ) looks similar to the Senegalo-Mauritanian Upwelling System (SMUS) used in Sylla et al. (2019) or the Mauritania-Senegalese Upwelling Region (MSUR) used in Vázquez et al. (2023). Please clarify what is the SMFZ compared to SMUS and MSUR.**

**REPLY:** The SMFZ is a "front" and more confined area than SMUS and MSUR and the upwelling should be one of factor to frontogenesis of the SMFZ (this has not been done yet, but Koseki et al., 2019 has done for Angola-Benguela Frontal Zone. A similar work will be possible in the future). Then, we modified the sentence. Please see line 27.

**L.26: "[...] one of the most pronounced oceanic frontal zones generated along the eastern boundary current system". Source?**

**REPLY:** Added Oettil et al. (2021).

**L.28: Please remove the second left bracket.**

**REPLY:** Done.

**L.36: Please remove the second left bracket.**

**REPLY:** Done.

**L.43: "[...] stronger Benguela Niño events compared to Dakar Niño events". Source? (Probably L.116-117).**

**REPLY:** Because there is no paper (as far as we know) for direct comparison between Beneguela and Dakar Niños, we omitted that part.

**L.47: Local or multi-fleet fishery? The former is certainly more sustainable, but also more sensitive than the latter. The latter has probably more impact on worldwide economy than the former. Please clarify.**

**REPLY:** In this study we do not specifiy which types of fleet and ships. Rather, our statement would be more general.

**L.64: "Sylla et al. (2022)" is missing in the references.**

**REPLY:** Added.

**L.65: "Resolution" instead of "Resoliution".**

**REPLY:** Corrected.

**L.86: Please explain what RCP8.5 is by adding what is said at L.138 (Which then can be shorten as "Under RCP8.5").**

**REPLY:** Added. Please see line 88.

**L.119-120: What is the source of the poor representation of coastal upwelling in ERA5?**

**REPLY:** This might be due to the relatively coarser resolution than ESA. Please see 123.

**L.120: "[...] and ERA5 has a warm bias (Vázquez et al., 2022)."**

**REPLY:** Corrected.

**L.134: "The meridional SST gradient greater than 0.5K/100km is shown in blue".**

**REPLY:** Corrected.

**L.134: Why between 21° and 16°W when the DNI is defined between 21° and 17°W? Is it a typo?**

**REPLY:** Corrected.

**L.135: "[...] respectively (bottom).".**

**REPLY:** "(bottom)" is for the SST standard deviation.

**L.135: "deviation" instead of "devitation".**

**REPLY:** Corrected.

**L.136: "kelvin" instead of "Kelvin".**

**REPLY:** Corrected.

**L.143: March is the peak phase of the Dakar Niño.**

**REPLY:** Corrected and modified following another reviewer. Please see lines 144-148.

**L.148: "isotherm" instead of "of temperature".**

**REPLY:** Corrected.

**L.165: "21°–17°W, 9°–14°N"**

**REPLY:** Corrected.

**L.167: "SST over 21°–17°W, 9°–14°N".**

**REPLY:** Corrected.

**L.168: How is the significance of the correlations evaluated. And for the wind stress, because there are two components (zonal and meridional), both can be significant, as well as only one over two. Does the figure only shows correlations when significant in both directions? Please clarify.**

**REPLY:** The significance of correlation is based on $p$-value ($p < 0.05$) as captioned. The vector with $u$ "or" $v$ is significant. We added this in the caption. Please see line 186-188.

**L.193: "section" instead of "seciton".**

**REPLY:** Corrected.

**L.196: "Niñas" instead of "Niña".**

**REPLY:** Corrected.

**L.210: "section" instead of "seciton".**

**REPLY:** Corrected.

**L.212: Is 16°W also a typo?**

**REPLY:** Corrected.

**L.253-254: "Composite anomalies of the 2m temperature during Dakar Niño (left) and Niña (right) events in ERA5 (top), ROM$_P$(middle), and ROM$_F$ (bottom) in March."**

**REPLY:** Corrected.

**L.277: Unclear what is the variability is referring to. Is it among warm (cold) events?. Is it in terms of number of warm (cold) events. Please rewrite and clarify.**

**REPLY:** Here, the variability is referred to SST inter-annual variability defined as Dakar Index. As the composite analysis suggests, Dakar Niña (cold SST anomalies) tend to intensify more. We repharsed the sentence. Please see lines 332.

**L.324-524: The references doesn't follow the Copernicus Publications guidelines. Please revise according to them.**

**REPLY:** Actually, we followed https://publications.copernicus.org/for_authors/manuscript_preparation.html and download the format file for Endnote for Copenicus Publication. After we submitted our manuscript was processed through technical check and then, it was sent to the editor and reviewers. Therefore, we assume that the style of reference is acceptable.

**Figure 3: In order to understand Fig.4, we need to know the mean state from January to May for the SST and the wind stress.**

**REPLY:** Thank you for the comment. We added this as new Fig.S3.

**Figure 5: The land mask should be in a different color than 0 (gray for example, similar to the mask in Fig. 7)**

**REPLY:** Corrected.

**Figure 6: Same as Fig. 5**

**REPLY:** Corrected.

**Figure 7: Is the standard deviation calculated for all the months of March in "Present" (ERA5 and ROM$_P$) and "Future" (ROM$_F$) periods? If it's the case, why not doing a composite with/without Dakar Niño/Niña?**

**REPLY:** Yes, the standard deviation in the plot includes all the months of March. In this study, we focus on how the Dakar Niño/Niña (extreme events of SST anomalies) will be changed under global warming. Therefore, we made a composite for Dakar Niño and Niña in the present and future climate to investigate which process can explain the changes in the events.

**References used in this review**

Cropper, T. E., Hanna, E., and Bigg, G. R.: Spatial and temporal seasonal trends in coastal upwelling off Northwest Africa, 1981–2012, Deep Sea Res. Part I Oceanogr. Res. Pap., 86, 94–111, https://doi.org/10.1016/j.dsr.2014.01.007, 2014.

Oettli, P., Morioka, Y., and Yamagata, T.: A regional climate mode discovered in the North Atlantic: Dakar Niño/Niña, Sci. Rep., 6, 18782, https://doi.org/10.1038/srep18782, 2016.

Oettli, P., Yuan, C., and Richter, I.: The other coastal Niño/Niña—the Benguela, California, and Dakar Niños/Niñas, in: Tropical and Extratropical Air-Sea Interactions, edited by: Behera, S. K., Elsevier, 237–266, https://doi.org/10.1016/B978-0-12-818156-0.00010-1, 2021.

Richter, I.: Climate model biases in the eastern tropical oceans: Causes, impacts and ways forward, WIREs Clim. Change, 6, 345–358, https://doi.org/10.1002/wcc.338, 2015.

Sydeman, W. J., García-Reyes, M., Schoeman, D. S., Rykaczewski, R. R., Thompson, S. A., Black, B. A., and Bograd, S. J.: Climate change and wind intensification in coastal upwelling ecosystems, Science, 345, 77–80, https://doi.org/10.1126/science.1251635, 2014.

Sylla, A., Mignot, J., Capet, X., and Gaye, A. T.: Weakening of the Senegalo–Mauritanian upwelling system under climate change, Clim Dyn, 53, 4447–4473, https://doi.org/10.1007/s00382-019-04797-y, 2019.

Vázquez, R., Parras-Berrocal, I. M., Koseki, S., Cabos, W., Sein, D. V., and Izquierdo, A.: Seasonality of coastal upwelling trends in the Mauritania-Senegalese region under RCP8.5 climate change scenario, Sci. Total Environ., 898, 166391, https://doi.org/10.1016/j.scitotenv.2023.166391, 2023.

Xiu, P., Chai, F., Curchitser, E. N., and Castruccio, F. S.: Future changes in coastal upwelling ecosystems with global warming: The case of the California Current System, Sci Rep, 8, 1–9, https://doi.org/10.1038/s41598-018-21247-7, 2018.

---

## Author Response (AR2)

**Reply to Reviewers' Comments**

**First Reviewer**

**Review for revised version of "Dakar Niño under global warming investigated by a high-resolution regionally coupled model" by Koseki et al.**

**I appreciate the authors' effort in addressing the issues raised in my previous review. I find the revised manuscript to be improved, but there are still a number of issues that need to be taken care of before publication, in particular regarding the clarity of the presentation and the processes driving Dakar Niños.**

Thank you very much for further careful reading and constructive comments. We revised the manuscript following the comments and we reply to the comments point-by-point as below. Please note that the track changes are shown by blue color in the revised manuscript.

**Major comments:**

**A) I find it still very hard to follow the manuscript and its argumentation. Please make sure that is always clear to the reader why certain analyses are done and what is shown in the plots before getting to what the results are. As an example, in the paragraph starting from line 190, it would be instructive to first state that in order to describe the evolution of Dakar Niños and Niñas and to evaluate how well it is simulated in the model and how it might change in the future (if that is the purpose of the figure - I am actually not sure!), correlation of the Dakar Niño index with the large scale wind stress and SST field are performed. Then you can get to where these correlations are high and what this means.**

**REPLY**: Thank you very much for this comment. We added more introductive descriptions on the plots of Figs. 5, 6, 7, and 8 for enhancing readability because we found that these plots lacked some explanation for the readers. Please read lines 195-196, 213-217, 227-229, and 249-250.

**As another example, for Fig. 6, there is an abrupt transition from "climatology of SST and wind stress is given" to "In ROMP, the significant positive**

correlation concentrates between the surface and 40m depth and decreases to 100m depth, which is about 0.4 (Fig. 6a)." Please state first what is looked at in Fig. 6 and why before getting to what one can see, i.e. that the Dakar Niño index is well correlated with the ocean temperature over the upper 40m of the water column.

**REPLY**: First, we deleted the sentence "climatology of SST and wind stress is given", which is not adequate there. As we replied to the previous comment, we added more descriptions before the plot of Fig.6. There, we refer to Fig.S4. Please see lines 214-217.

**B) Related to point A, the discussion of the correlation figures is rather confusing. Please note that correlations can be high or low and variables can be highly/strongly or weakly correlated, but not "deeply".**

**In particular for Fig. 6, I cannot make sense of what is suggested here. The plots indicate that there are higher subsurface temperatures during Dakar Niños and weaker upwelling. To me, this implies that, in the model, weaker winds lead to reduced upwelling and turbulent mixing, warming the subsurface and subsequently the surface ocean. This process appears to get more important in the future. It does not mean that "Dakar Niños would influence deeper ocean in the future" (as stated in lines 207/208).**

**REPLY**: Thank you very much for the insightful discussion. Here, "deeply" we used means that the higher (or, significant) correlation with Dakar Index can be seen at deeper layer in ROM$_F$ than ROM$_P$. Because Fig.6 shows just correlation, the information does not indicate any strength of variability. However, we agree that such correlation pattern does not mean **"Dakar Niños would influence deeper ocean in the future"**, and so we rephrased that part. Please see lines 216-222.

**C) Again related to point A, the newly added heat budget calculation appears rather disconnected from the rest of the manuscript. While in lines 235/236, it is stated that "According to Oettli et al. (2016), the Dakar Niño is mainly driven by changes in alongshore local surface wind", this section (from line 294) starts with stating that "According to Oettli et al. (2016), surface heat flux is responsible for generating Dakar Niño events". This needs at least some more context and explanation as the previous section has not mentioned whether the wind is important because of its impact on latent heat flux. By showing vertical velocities and subsurface temperatures, it is rather implied that it is the effect of wind changes on upwelling that are important.**

**REPLY**: Thank you very much for the insightful discussion. First, the expression "Dakar Niño is mainly driven by changes in alongshore local surface wind" is not correct and we changed "driven" to "associated with". Please see line 249.

During this revision, we re-visited the heat budget of the previous version and we found something wrong in the calculation: the surface heat flux was underestimated in the code. So, first please let us correct the discussion on the heat budget. Here, we show a plot of the corrected heat budget as Fig.R1,

[Figure]

**Fig.R3**. Corrected version of composite differences in the heat budget in the box of 9N-14N and 20W-17W between the Dakar Niños and Dakar Niñas for the ROM simulations. Black is for the present (ROM$_P$) and grey is for the future climate (ROM$_F$).

As Fig. R1 shows, the ROM simulations also show a vital role of surface heat flux as Oettli et al. (2016). In March, the heat flux intensifies more in the future and this strengthened heat flux can explain the stronger Dakar Niño/Niña. This larger heat flux anomaly is due to the stronger surface wind anomaly as Fig. 8 shows (correlation of heat flux and meridional wind stress is 0.88 over the Dakar Index box in March). In addition, mixed layer depth anomaly is also larger in the future, and this can help enhancing surface heat flux contribution as we described in the previous manuscript.

However, in ROM$_P$, the heat flux anomaly for the Dakar Niño events seems to be underestimated especially in January and February. Oettli et al. (2016) showed that shortwave radiation anomaly is a main contributor to the positive heat flux anomaly inducing Dakar Niño, but it is not discussed what generates the shortwave radiation anomaly in detais. There are two possible internal factors: cloud and aerosol. Especially, the focusing region is in the vicinity of the Sahara where dust emission is the largest. As Chen et al. (2021), the dust from the Sahara is quite important in surface heat budget in the north tropical Atlantic and they showed a cooling effect of dust on SST. Because ROM implements "climatological" dust forcing (Pietikäinen et al., 2012, we cited this in the revised manuscript), the heat flux anomalies may be not well represented inducing the Dakar Niños. It is very insightful to investigate how dust anomalies can induce shortwave anomalies and consequently, SST anomalies, but it is out of scope of this study. This point should be made in one of future works.

As another reviewer suggests, we also computed the heat budget for ORAS5 given in Fig. R2. The events of Dakar Niño and Niña are the same between ERA5 and ORAS5 because Dakar Indices give almost identical characteristics as shown in Fig. R3.

[Figure]

**Fig.R2**. Composite differences in the heat budget in the box of 9N-14N and 20W-17W between the Dakar Niños and Dakar Niñas for ORAS5.

[Figure]

**Fig.R3**. Dakar Index for (black) ERA5 and (red) ORAS5. Orange and blue dots denote Dakar Niño and Niña events in ERA5 as given in Fig. 4 of the manuscript.

As Oettli et al. (2016), ORAS5 indicates that net heat flux is responsible for Dakar Niño in 1-2 months advance. Horizontal advection is also comparably important to the Dakar Niño. The magnitude of the estimated horizontal advection is roughly consistent with that of ROM$_P$(please see Fig.R3). Between ORAS5 and ROMP, the horizontal advection is comparable, and heat flux shows a difference in magnitude: ROM simulation underestimates the heat flux partially because shortwave radiation anomalies due to dust could be represented as we describe above.

Following this correction and additional analysis, we modified significantly our discussion and conclusion. Please see lines 22-25, 329-350, and 371-373. We added Fig.R2 and R3 as new Fig. S6 and Fig.R1 are replaced with Fig.11.

Regarding the isolation of heat budget section, we added some texts in the introduction to guide the readers to heat budget analysis. Please see lines 41-42, and 74-76.

**Reference**:

Chen, S.-H., Huang, C.-C., Kuo, Y.-C., Tseng, Y.-H., et al. 2021: Impacts of Saharan Mineral Dust on Air-Sea Interaction over North Atlantic Ocean Using a Fully Coupled Regional Model. JGR-Atmosphere, https://doi.org/10.1029/2020JD033586.

Pietkäinen, J.P., O´Donnell, D., Teichmann, C., Karstens, U., et al., 2012. The regional aerosol-climate model REMO-HAM, Geosci. Model Dev., 5, 1323-1339, 10.5194/gmd-5-1323-2012.

**Surprisingly, the equation used for the heat budget does not even contain a term for the surface heat fluxes that, however, show up in Fig. 11 without ever being introduced before. This Figure then suggests that it is the horizontal advection that is mainly driving Dakar Niños and Niños, another process that has not really been introduced so far.**

**REPLY**: Thank you very much for raising this point. This is completely our mistake. We added net hat flux term in the heat budget equation and some descriptions. Please see lines 317 and 321-322.

**To address these major comments, I would advice the authors to first show that Dakar Niños are related to the wind field (Fig. 5) and that the wind variability is intensifying in the scenario simulation (Fig. 8), explaining the increase in SST variability. Then there could be a section on the different mechanisms that are related to wind variability, namely upwelling, advection and latent heat flux and how each of them is changing.**

**REPLY**: Thank you very much for this constructive comment. Our philosophy is that first we would like to show the future change of SST variability (Fig.2) so that we could draw attention from readers. Showing sequentially the change of other oceanic properties and connection to SST variability (Figs.5-7) could extend the overview of change in Dakar Niño. Then, in Section 4, we discuss in detail why Dakar Niño is amplified focusing on winds and heat budget. As we replied to Comment (A), we added more introductive descriptions on the plots, we would suppose that the flow of story now increases readability. However, since Section 4.1 becomes a long section after the revision, we divided wind change and heat budget to Section 4.1 and 4.2.

Minor points:

**1) I don't understand what is meant by lines 17/18 in the abstract (see also major comment B above).**

**REPLY**: As we replied to the Major Comment B, that means that temperature and vertical velocity are correlated with SST at deeper layer in the future. So, we rephased that part of abstract. Please see lines 18-19.

**2) Instead of the current Figure 1, I would suggest to show a map for the coupled domain with SST standard deviation in shading and mean SST overlaid in contours as well as the box indicating the Dakar Niño region. Then this area of high SST variability can be referred to early in the manuscript.**

**REPLY**: Thank you very much for the comment. We re-plotted Fig. 1 following the comment and referred the Fig.1 in the introduction as well. Please see the new Fig.1 and lines 28 and 40.

**3) Related to major comment A above, the transition in line 160 is not very clear. Please state explicitly that you are now referring to future changes.**

**REPLY**: We rephrased that part for better connectivity. Please see lines 163-164.

**4) In the discussion of the Dakar Niño index time series (lines 172 to 177), please also refer to the change in standard deviation visible in Fig. S1.**

**REPLY**: We added it. Please see line 181-182.

**Specific comments:**

**- line 21: "more important" instead of "more explainable"**

**REPLY**: Replaced.

**- line 25: "Climatologically" instead of ""From a climatological aspect"**

**REPLY**: Replaced.

**- line 29: "northern boundary" instead of "northern end"**

**REPLY**: Replaced.

**- line 30: "joins" (typo), "North Equatorial Current"**

**REPLY**: This is not a typo, but that is what we meant.

**- line 59: "studies" or "analysis" instead of "surveys"**

**REPLY**: Replaced.

**- line 59: add recent study on future evolution of Benguela Niños by Prigent et al., (2024)**

**REPLY**: Here, we mention "equatorial" Atlantic variability while Prigent et al. studies Benguela Niño, which is not equatorial. Instead, we cited Prigent in the conclusion section.

**- line 76: "configuration … is" or "configurations … are"**

**REPLY**: Replaced.

**- line 77: "… atmospheric component, namely the limited-area…"**

**REPLY**: Done.

**- line 78: "… and a global oceanic component, which is the Max-Planck Institute Ocean Model…"**

**REPLY**: Done.

**- line 85: "this" instead of "those"**

**REPLY**: Replaced.

**- line 99: "shading shows" instead of "shade show" Is the topography height actually relevant for the study? Maybe it would be more instructive to show SST instead.**

**REPLY**: As we replied to minor comment #2, we replotted SST in Fig.1.

**- line 110/111: Not sure what is meant by "The steep SST gradient is consistent with the SST seasonal cycle"**

**- line 126: What is meant by "in a whole year"? "during the whole year" or "for all calendar months" maybe?**

**REPLY**: Corrected to "during the whole year" as suggested.

**- line 136: "absolute" (typo)**

**REPLY**: Corrected.

**- line 137: "meridional" (typo)**

**REPLY**: Corrected.

**- line 141/142: This is not a full sentence and it is not clear what is meant. Please rephrase.**

**REPLY**: Rephrased it.

**- line 146: "focus on"**

**REPLY**: Corrected.

**- line 148: Please remove "for fairness comparison with observations"**

**REPLY**: Removed.

**- line 149: "averaged over 9ºN - 14ºN"**

**REPLY**: Added.

**- line 152: "The yellow box indicates the Dakar Niño index region."**

**REPLY**: Corrected.

**- line 172: "in ERA5, there are" instead of "ERA5 counts"**

**REPLY**: Corrected.

**- line 197: Part of the sentence seems to be missing.**

**REPLY**: Corrected.

**- line 198: It's the connection between the Dakar Niño index and the wind field that is not well simulated.**

**REPLY**: Corrected.

**- line 204: Why is the climatology shown here? What is the connection to the preceding or following text?**

**REPLY**: As we replied to Major Comment A, we corrected the connectivity and the texts on Fig. S4 are given in the explanation of Fig. 5. Please see lines 214-215.

**- line 253/254: "more intensely than the ocean"**

**REPLY**: Corrected.

**- line 266: "In the climatology"**

**REPLY**: Corrected.

**- line 297 and in many following sentences: "ocean mixed layer" instead of "ocean mixing layer"**

**REPLY**: Corrected.

**- line 338: "in contrast to" instead of "in discrepancy against"**

**REPLY**: Corrected.

**Second reviewer**

**I am not satisfied with the revision and the responses provided by the authors. Particularly, the authors continue to only focus on the alongshore wind, while the mechanism behind the Dakar Niño/Niña is more complex, as already highlighted in the review.**

**REPLY**: Thank you very for the comment. In the first revision, we elaborated our discussion by adding the heat budget analysis and we suggested an importance of horizontal advection for Dakar Niño. In this revision, we added a heat budget analysis for reanalysis data (please see our reply to a major comment as below) and we recognized that surface heat flux is responsible for Dakar Niño, but our ROM simulation appears to underestimate the contribution of heat flux. According to Oettli et al. (2016), shortwave radiation plays a vital role in inducing the Dakar Niños. So, it is likely that our ROM fails to represent such shortwave radiation anomaly and it is important to investigate what makes the shortwave radiation variability, probably, by cloud and/or dust from the Sahara. This point is quite important and insightful, but at this moment, it is out of scope and will be explored in the future.

**To the credit of the authors, they have performed some heat budget (as suggested in the review) but the results are quite different from those found in Oettli et al. (2016), which is fine. But the differences are quite drastic and seem to invalidate previous studies. This must be thoroughly discussed in the manuscript. Also, the heat budget is not performed on observation/reanalysis. This should be done and shown at least as supporting information, to compare with ROMp heat budget (Figure 11).**

**REPLY**: Thank you very for the comment. Yes, heat budget analysis for reanalysis data is important to elaborate our discussion. Therefore, we performed a heat budget analysis with ORAS5 (approximately 0.25°x0.25° resolution). Please note that ORAS5 does NOT provide vertical velocity, and we could not estimate vertical advection. First, we checked the Dakar Niño/Niña events in ORAS5 comparing to ERA5 as shown Fig.R1.

[Figure]

**Fig.R1**. Dakar Index for (black) ERA5 and (red) ORAS5. Orange and blue dots denote Dakar Niño and Niña events in ERA5 as given in Fig. 4 of the manuscript.

Actually, ORAS5 performance is quite similar to ERA5 and then, we can regard that the Dakar events of ORAS5 are same as ERA5. Then, we calculated the heat budget of ORAS5 in the events as given in Fig.R2.

[Figure]

**Fig.R2**. Composite differences in the heat budget in the box of 9N-14N and 20W-17W between the Dakar Niños and Dakar Niñas for ORAS5.

As Oettli et al. (2016), ORAS5 indicates that net heat flux is responsible for Dakar Niño in 1-2 months advance. Interestingly, horizontal advection is also comparably important to the Dakar Niño. The magnitude of the estimated horizontal advection is roughly consistent with that of $ROM_P$ (please see Fig.R3).

[Figure]

**Fig.R3**. Corrected version of composite differences in the heat budget in the box of 9N-14N and 20W-17W between the Dakar Niños and Dakar Niñas for the ROM simulations. Black is for the present (ROM$_P$) and grey is for the future climate (ROM$_F$).

During this revision, we re-visited our previous heat budget for the ROM simulations and we found that the computation of heat flux was wrong. Here, we show a correct heat budget of the ROM simulations in Fig.R3.

As Fig. R3 shows, the ROM simulations also show a vital role of surface heat flux as Oettli et al. (2016) and ORAS5 analysis. In March, the heat flux intensifies more in the future and this strengthened heat flux can explain the stronger Dakar Niño/Niña. This larger heat flux anomaly is due to the stronger surface wind anomaly as Fig. 8 shows (correlation of heat flux and meridional wind stress is 0.88 over the Dakar Index box in March). In addition, mixed layer depth anomaly is also larger in the future and this can help enhancing surface heat flux contribution as we described in the previous manuscript.

However, in ROM$_P$, the heat flux anomaly for the Dakar Niño events seems to be underestimated especially in January and February compared to ORAS5. Oettli et al. (2016) showed that shortwave radiation anomaly is a main contributor to the positive heat flux anomaly inducing Dakar Niño, but it is not discussed what generates the shortwave radiation anomaly in details. There are two possible internal factors: cloud and aerosol. Especially, the focusing region is in the vicinity of the Sahara where dust emission is the largest. As Chen et al. (2021), the dust from the Sahara is quite important in surface heat budget in the north tropical Atlantic and they showed a cooling effect of dust on SST. Because ROM implements "climatological" dust forcing (Pietikäinen et al., 2012, we cited this in the revised manuscript), the heat flux anomalies may be not well represented inducing the Dakar Niños. It is very insightful to investigate how dust anomalies can induce shortwave anomalies and consequently, SST anomalies, but it is out of scope of this study. This point should be made in one of future works.

We added Figs. R1 and R2 as supplemental information as new Fig. S6 and corresponding descriptions. Moreover, we corrected Fig. 11 and our conclusion in line with correction of heat budget of ROM. Please see lines 22-25, 329-350, and 371-373. Fig.R3 is replaced with Fig.11.

**Reference**:

Chen, S.-H., Huang, C.-C., Kuo, Y.-C., Tseng, Y.-H., et al. 2021: Impacts of Saharan Mineral Dust on Air-Sea Interaction over North Atlantic Ocean Using a Fully Coupled Regional Model. JGR-Atmosphere, https://doi.org/10.1029/2020JD033586.

Pietkäinen, J.P., O´Donnell, D., Teichmann, C., Karstens, U., et al., 2012. The regional aerosol-climate model REMO-HAM, Geosci. Model Dev., 5, 1323-1339, 10.5194/gmd-5-1323-2012.

**Also, the Figure 4 is interesting, but not really discussed. We can see that the years of Dakar Niño/Niña are more or less the same than in Oettli et al. (2016), but using ERA5. Which is an important result. But with ROMp, the results are different. A few years of observed Dakar Niño/Niña are detected (1985, 2003 for example). But there are also important discrepancies that must be thoroughly discussed. What explains the strong 1999 Dakar Niño in ROMp, when it was a strong Dakar Niña? Same applies to 2009. Also, 1988 and 1989 were neutral, but ROMp simulated 2 consecutive Dakar Niños. Finally there are 6 consecutive Dakar Niñas between 2002 and 2007. This doesn't argue in favor of ROM**.

**REPLY**: Because our ROM simulations are forced by MPI-ESM "historical" and RCP585 scenario runs from 1950 to 2100 (already mentioned in Section 2, please see lines 88-91), we do not expect that the Dakar Niño/Niña events in $ROM_P$ occur in line with ERA5. We added one sentence to notify the setting of ROMP in the Section 3 as well. Please see lines 178-180.

Regarding the consistency with Oettli et al. (2016), we also added some descriptions to justify our results of ERA5. Please see lines 175-176.

**It should be noted thath there are a lot of issues with the citations throughout the text (i.e, open parenthesis not correctly placed). This makes the text difficult to read**.

**REPLY**: Thank you very much for the comment. We checked carefully the manuscript and corrected the wrong parentheses throughout the manuscript.

**I therefore recommend major revisions.**

**Some comments (non-exhaustive list):**

**L.145: "To assess the Dakar Niño...". Not sure what the authors mean here. Is it to assess the existence of the phenomenon? To assess the simulation of Dakar Niño in ROM? Please elaborate.**

**REPLY**: This expression seems a bit confusing and therefore, we deleted "to assess the Dakar Niño".

**L.235: "[...] the Dakar Nino is driven mainly by changes in alongshore local surface wind". No, it is not, as already said in the review.**

**REPLY**: We changed "driven by" to "associated with" as Oettli et al. (2016) say. Please see line 249-250.

**L.312-315: Does it mean that ROM generates abnormal coastal SST warming/cooling by horizontal advection? How about ERA5 (hence the mandatory heat budget). Could this explain the differences between ERA5 and ROMp described above?**

**REPLY**: As we replied to the major comment #1, we calculated the heat budget for ORAS5 reanalysis (Dakar Niño/Niña events are almost identical with ERA5). Actually, ORAS5 result is similar to Oettli et al. (2016) showing an important role of surface net heat flux and importance of horizontal advection. On the other hand, $ROM_P$ underestimates the surface heat flux anomaly. However, horizontal advection in ORAS5 and $ROM_P$ are roughly consistent.

**L.287: "The SLP anomaly gradient runs along..." The gradient is across. It is part of the generation process.**

**REPLY**: Corrected. Please see line 301.

---

## Author Response (AR3)

**Reply to Reviewer#1**

**Review for revised version of "Dakar Niño under global warming investigated
by a high-resolution regionally coupled model" by Koseki et al.**

**After the second revision, I find the manuscript to be substantially improved, especially regarding the readability. However, the discussion of the relevant processes still needs more work.**

Thank you very much for further careful reading and constructive comments. We revised the manuscript in response to the comments and provide point-by-point replies to each comment below. Please note that the track changes are highlighted in blue in the revised manuscript.

**Major comments:**

**There are several processes for the generation of Dakar Niños and Niñas discussed in the paper, namely local winds, upwelling (although this is somehow only touched on), surface heat fluxes and horizontal advection. It needs to become clear that and how these processes are connected. This starts as early as lines 40/41 in the introduction and continues throughout the manuscript.**
**Changes in the local winds can drive changes in upwelling, horizontal advection, latent heat loss and mixed layer depth but there are also components not (directly) related to local winds such as temperature gradients and shortwave radiation. At the moment, it is not clear to me what the main role of the winds is for generating SST anomalies and which part of the net surface heat flux is actually important.**

**REPLY**: Yes, local winds influence all terms of heat budget as shown in our heat budget analysis. Due to changes in the local wind variability between the present and future climate, we have already shown that vertical advection and horizontal advection terms are identically (but, secondarily) important in strengthening the SST variability in the focusing area. As shown in Fig. 5 and Oettli et al. (2016) shows, SST variability and meridional wind variability exhibit strong coherence, suggesting that SST variability is associated with wind variability. Additionally, as Oettli et al. (2016) discusses, surface heat flux also plays an important role in

inducing the SST variability. According to Oettli et al. (2016), shortwave radiation is the dominant factor, with latent heat flux being secondarily important.

As we noted in the previous revision, Oettli et al. (2016) does not specify the factors controlling the shortwave radiation variability (perhaps, cloud and/or dust from the Sahara). This remains an open question for future research. In response to the major comment below, we have divided the surface heat flux.

**Specifically, I would suggest to start the heat budget analysis (Section 4.2) by saying that**
**a) variations in local winds are important for the generation of Dakar Niños and Niños (as shown in the previous section)**
**b) these wind variations can be connected to a number of processes, namely upwelling, ocean currents, and latent heat loss**
**c) that you perform a mixed layer heat budget analysis to determine which of these is most important**
**It is a bit confusing to start the section by pointing out the importance of heat fluxes as found by Oettli et al. (2016).**

**REPLY**: Thank you very much for the suggestion. Now, we reconstructed the introductive part of the heat budget analysis. Please see lines 314-315. To highlight the difference and similarity with Oettli et al. (2016), we have retained the sentence at 316.

**Given the high correspondence between SSTs and subsurface temperatures over at least the upper 100m of the water column during Dakar Niño and Niña events (Fig. 6, 7), I find it very hard to believe that vertical processes are not important for the generation of the events as stated later with respect to the heat budget analysis. I rather suspect that a large part of the vertical term (especially mixing) is buried in the residual which is probably quite large. How much of dT/dt do the advection and heat flux terms actually account for, i.e. how large is the residual term in the heat budget?**

**REPLY**: We agree that the mixing and entrainment processes are important in inducing the SST variability. According to Oettli et al. (2016), entrainment is also important factor for the SST variability. However, our available data is limited to monthly intervals and there is no output of entrainment and higher order variables. We have added this at lines 316-317 and 327.

Since entrainment and mixing are more enhanced around mixed-layer depth or thermocline depth, deeper connection with the Dakar Index does not necessarily mean that entrainment and mixing are stronger in the future than in the present. In Fig. R1 and R2, the Dt/dt estimated from monthly data and the sum of the explicit heat budget analysis (horizontal advection, vertical advection, and heat flux) are presented. Please note that R1 shows the DT/Dt for the difference of Feb-Jan, Mar-Feb, Apr-Mar, and May-Apr. From these plots, it seems that the residual term could be positive, but its magnitude is smaller than horizontal advection and surface heat flux from January to February. In March, the residual term can be negative. Therefore, in the ROM simulations, the residual term is not a main source for inducing Dakar Niños. The residual term includes not only entrainment, but also more higher-order terms. Again, accorgint to Oettli et al. (2016), the entrainment is a secondarily important role and we can suggest that the entrainment alone does not explain the enhancement of Dakar Niño under global climate in our simulations. However, this analysis is based on monthly-mean data, which is too coarse to capture the full complexity of the heat budget. Therefore, we will need to analyze high-frequency data explicitly. For those reasons, in this paper, we focus on the heat budget that we showed in the previous revision.

[Figure]

**FigR1.** The SST tendency difference between Dakar Niño and Niña, D(SST)/dt (K/day) is estimated from monthly SST composite.

[Figure]

**FigR2.** Monthly composite difference between Dakar Niño and Niña for sum of the explicit terms of the ehat budget (horizontal advection, vertical advection, and surface heat flux). The unit is K/day.

**I find the discussion of the heat budget analysis (Section 4.2) partly confusing and I am not sure if there maybe was a mix-up of the legends. If the black lines show ROMp, horizontal advection appears much more important than the surface heat fluxes (in contrast to what is stated in line 330). The black lines also show a stronger heat flux damping in April than the grey line (in contrast to what is stated in lines 358/359).**

**REPLY**: Probably, our explanation was a bit confusing. Between January and March, surface heat flux (dot lined) and horizontal advection (solid line) in ROMP (black) are roughly comparable while the peak timing is different. We rephrased that part. Please see lines Regarding lines 358/359, yes, that was wrong. We deleted that sentence.

**Also the components of the net surface heat flux need to be looked at individually. If shortwave radiation would be dominant (as indicated in line 336), how does this fit with the importance of the local winds?**

**REPLY**: Thank you very much for raising this point. As shown in Fig. R3, we have divided the surface heat flux into 4 components. First, regarding the line 336, which concerns about Oettli et al. (2016), our ROM simulation might underestimate the shortwave radiation contribution due to the poor representation of aerosol (which are fixed to climatology).

[Figure]

**Figure R3:** Monthly time series of lag-composite difference of latent heat flux (red), sensible heat flux (blue), longwave radiation (green), and shortwave radiation (magenta) for ROM$_P$ (solid) and ROM$_F$ (dashed) with the Dakar Index box. The unit is W/m$^2$.

In the present climate (sold lines in Fig.R3), latent heat flux is dominant from January to February, with shortwave radiation also playing an important role in February. Sensible and longwave are quite minor. This timing aligns with the results of Oettli et al. (2016). However, as we mentioned above, their result shows the dominance of shortwave radiation, while our result shows a more modest shortwave contribution, partially due to the poor representation of aerosol variability.

In the future climate (the dashed lines in Fig. R3), latent heat flux is responsible for enhancement of surface heat flux in March, with shortwave radiation also contributing to the future increase of the heat flux. At present, the processes cause the shortwave anomaly and the enhancement in the future are unclear. Possible processes are cloud and aerosol changes, but this is out of scope of our study and will be investigated in future work.

We added Fig.R3 as Fig.12 and descriptions on this discussion. Please see lines 356-372, 24, and 387. Please note that some texts on dust and the references are also moved to this part so that the flow of discussion become fluent.

**Minor points:**

**a) Please indicate units in Figure 1.**

REPLY: Added in the caption.

**b) In lines 107 to 110 the seasonal migration of the SST front is described as a result of the seasonal cycle of the Canary Current and changes in the**

**location of upwelling. Doesn't it also reflect the seasonality of the upwelling strength?**

**REPLY**: We agree. The changes primarly affect the seasonality of upwelling rather than the Canary Current. We have modified the text and added that this shift is associated with the migration of the ITCZ. The updated text now reads as follows Please see lines 107-110.

"This seasonal meridional migration of the SST front is linked to the seasonal cycle of the Canary upwelling system (Cropper et al., 2014; Pardo et al., 2011; Sylla et al., 2019) and is associated with the northward migration of the Intertropical convergence zone in the summer months, which displaces the surface water masses meridionally."

**c) CMIP models are coupled climate models, not Earth system models which typically include more components such as biology (e.g. line 134)**

**REPLY**: Corrected.

**d) In Figure S2(b) there is some weird color shading at the shelf.**

**REPLY**: Corrected.

**e) In Figure 4, please use the same y-axis for all subplots.**

**REPLY**: Corrected.

**f) I noticed some curious clustering of events in ROMp (Fig. 4b) with three consecutive Dakar Niños followed by five consecutive Dakar Niñas. Any idea what this might be related to?**

**REPLY**: At this moment, we do not have any clear explanation. However, consecutive events like these might suggest longer-time scale climate variability, such as Atlantic Multi-Decadal Oscillation (AMO). While AMO is out of scope in this study, its influence could be one of future studies on Dakar Niño.

**g) Related to my major comments above, the relation between upwelling, subsurface temperatures and SSTs needs to be explained before getting to Figure 5.**

**REPLY**: Before Fig. 5, we have only presented the connection between SST and surface wind connection as in Fig. 5 without showing any vertical structure. Therefore, we would think that mentioning SST and subsurface temperature before Fig.5, it might disrupt the flow of the story.

**h) Somewhere in the discussion of Fig. 5, please state that that this means that Dakar Niños are related to weakened Northeasterly trade winds.**

**REPLY**: Thank you for the suggestion. We added some texts. Please see line 201.

**i) The word "penetrates" in line 234 suggests that the warming comes from the surface and is transported downward. Is this the case? It could also be the other way around - a subsurface anomaly that is impacting the surface by vertical advection and mixing.**

**REPLY**: We agree. Therefore, that changed to "is detected" because there is no evidence of water penetration from the surface to the subsurface.

**j) To compare the heat budget analysis from the model to ORA-S5 reanalysis, vertical velocities can be calculated from the divergence of the horizontal velocity field (as done in many studies).**

**REPLY**: We agree. However, the ORAS5 analysis is just a supplement suggested by another reviewer and the result of re-analysis is already provided by Oettli et al. (2016). The ORAS5 is not our main purpose of this study. In addition, while many studies take this approach, using monthly-mean data and calculation from the divergence might cause further uncertainty in the result. Therefore, we prefer to show only the two explicit components of the ORAS5.

**k) Explain how variations in mixed layer depth are related to the importance of surface heat fluxes (line 356/357).**

**REPLY**: The heating rate from surface heat flux is divided by the depth of mixed layer $D$, meaning that thinner layer during Dakar Niño compared to Niña, can amplify the warming effect of the surface heat flux. We have added this explanation. Please see lines 353-354.

**l) In the Discussion, Chang et al. (2023) is cited. Please comment on how their results align with what you find.**

**REPLY**: In the two sentences preceding the citation of Chang et al. (2023), we described the differences in future response between Benguela Niño (Prigent et al., 2023) and Dakar Niño (our findings). There, we also mention importance of comparative studies between southern and northern upwelling regions. Chang et al. (2023) is one of the examples showing the difference between south and north, supporting our suggestion for future comparison studies. In addition, their study is about climatology of upwelling, not about inter-annual variability.

**Specific comments:**
**- line 20: "wind stress, which is"**

**REPLY**: Corrected.

**- line 29: "The cold water of the southward flowing Canary Current and the..."**

**REPLY**: Corrected.

**- line 86: There is no red rectangular in Fig. 1.**

**REPLY**: Changed to "yellow".

**- line 101: "panel"**

**REPLY**: Corrected.

**- line 146: "identical" instead of "identically".**

**REPLY**: Corrected.

**- line 147: Please also refer to Fig. S1.**
**REPLY**: Added.

**- line 149: "enhancement" instead of "reinforcement"**
**REPLY**: Corrected.

**- line 163: no comma after especially**
**REPLY**: Deleted.

**- line 171/172: either "we investigate" or "are investigated", not both**
**REPLY**: Deleted.

**- line 176: "agrees well" or "is consistent with"**
**REPLY**: Corrected.

**- line 177/178: Please specify the length of the time series again here.**
**REPLY**: Done.

**- line 195: "using a reanalysis product" (maybe specify which product)**
**REPLY**: Done.

**- line 195: "Dakar Niños are…"**

**REPLY**: Done.

**- line 197: Please rephrase, e.g. "In January, SST anomalies from ERA5 averaged over the Dakar Niño box are positively correlated with SSTs along the west African coast as well as southwesterly surface wind anomalies. The correlation strengthens in March which is the peak of the event."**

**REPLY**: Rephrased.

**- line 200: "significant" or "pronounced" instead of "more dominant"**
**REPLY**: Corrected.

**- line 201, 210: "The area positively correlated with SST shifts westward…"**
**REPLY**: Corrected.

**- line 214: A word is missing after "coastal".**

**REPLY**: Added "Ekman divergence"

**- line 218/219: "decreases to 0.4 at 100m depth (Fig. 6a).**
**REPLY**: Corrected.

**- line 219: "deeper in the water column" instead of "more deeply"**
  **REPLY**: Corrected.

**- line 248: The increase from ROMp to ROMf is actually strongest in April.**
  **REPLY**: Changed to "March to April".

**- line 259: "In contrast" instead of "Inversely"**

**REPLY**: Corrected.

**- line 260: I would say that it actually decreases.**
  **REPLY**: Corrected.

**- line 274: "becomes" (typo)**
  **REPLY**: Corrected.

**- line 275: "of the cool anomaly"**
  **REPLY**: Added.

**- line 276: "the land-surface"**
  **REPLY**: Corrected.

**- line 276 and 278: "towards the west" instead of "more westward"**
  **REPLY**: Corrected.

**- line 279: A word is missing after "much"**
  **REPLY**: Corrected to "larger".

**- line 289: "can also be seen"**

**REPLY**: Corrected.

**- line 290: "the SLP anomalies show"**
  **REPLY**: Corrected.

**- line 293: "in particular in the case of…"**
 **REPLY**: Corrected.

**- line 311: "investigate" or "examine" instead of "question"**
 **REPLY**: Corrected.

**- line 314/315: either "we considered" or "is estimated", not both**
 **REPLY**: Deleted "we consider"

**- line 319: "the bottom of the ocean mixed layer"**
 **REPLY**: Corrected.

**- line 320: "and just below"**
 **REPLY**: Corrected

**- line 321: "Q is the net…"**
 **REPLY**: Corrected.

**- line 324: "missing" instead of "missed"**
 **REPLY**: Corrected.

**- line 347: It's not clear what "almost identical" is referring to. Present and future? Please specify.**
 **REPLY**: This is both of advection term. Here, we added "the magnitude of enhancement in the future"

**- line 349/350: What does "remains secondary" mean? Does this refer to the future simulation?**
 **REPLY**: This is not correct explanation. We deleted that sentence.

**- line 357/358: What is meant by "more persisting"?**
 **REPLY**: Here we meant SST anomaly associated with Dakar Niño. As Fig.5 shows, the SST anomaly (correlation with Dakar Index) is still high value in April. Because of more heating in March in the future, the SST anomaly can survive in April in $ROM_F$ than $ROM_P$.

**- line 364: "an intensification of interannual SST variability in the…"**
 **REPLY**: Corrected.

**- line 365: "increase in the amplitude of…"**
 **REPLY**: Corrected.

**- line 367: "is in contrast"**
 **REPLY**: Corrected.

**- line 369: "coastal" instead of "coast"**
 **REPLY**: Corrected.

**- line 381: A part of the sentence seems to be missing.**
 **REPLY**: "that" is removed and changed to "teleconnections"

**- line 385: "modes such as"**
 **REPLY**: Corrected.

**- line 389: please rephrase "will be desired"**
 **REPLY**: Changed to "will be necessary"

**- line 399: I guess "performed" is meant (instead of demonstrated)**

**REPLY**: Corrected.